# PolarMix: A General Data Augmentation Technique for LiDAR Point Clouds

**Aoran Xiao[1], Jiaxing Huang[1], Dayan Guan[2], Kaiwen Cui[1], Shijian Lu[1],\* Ling Shao[3]**

[1]School of Computer Science and Engineering, Nanyang Technological University
[2]Mohamed bin Zayed University of Artificial Intelligence [3]Terminus Group, China
{aoran.xiao, jiaxing.huang, kaiwen.cui, shijian.lu}@ntu.edu.sg
dayan.guan@mbzuai.ac.ae, ling.shao@ieee.org

## Abstract

LiDAR point clouds, which are usually scanned by rotating LiDAR sensors continuously, capture precise geometry of the surrounding environment and are crucial to many autonomous detection and navigation tasks. Though many 3D deep architectures have been developed, efficient collection and annotation of large amounts of point clouds remain one major challenge in the analytics and understanding of point cloud data. This paper presents *PolarMix*, a point cloud augmentation technique that is simple and generic but can mitigate the data constraint effectively across different perception tasks and scenarios. PolarMix enriches point cloud distributions and preserves point cloud fidelity via two cross-scan augmentation strategies that cut, edit, and mix point clouds along the scanning direction. The first is scene-level swapping which exchanges point cloud sectors of two LiDAR scans that are cut along the azimuth axis. The second is instance-level rotation and paste which crops point instances from one LiDAR scan, rotates them by multiple angles (to create multiple copies), and paste the rotated point instances into other scans. Extensive experiments show that PolarMix achieves superior performance consistently across different perception tasks and scenarios. In addition, it can work as a plug-and-play for various 3D deep architectures and also performs well for unsupervised domain adaptation. Code is available at https://github.com/xiaoaoran/polarmix

## 1 Introduction

In the past decade, LiDAR sensors have been increasingly employed in various perception related applications such as autonomous driving. They provide accurate and robust depth sensing of the surrounding environments which is crucial for scene understanding for autonomous navigation indoors and outdoors. With the recent advance of deep neural networks (DNNs), point cloud understanding has achieved significant progress in various perception tasks such as semantic segmentation [47, 16, 41, 53, 26, 27] and object detection [23, 48, 9]. On the other hand, training reliable DNN models requires large amount of well-annotated training data, whereas collecting and annotating large amounts of point clouds is often laborious, time-consuming, and has poor scalability across tasks and domains. This has become one major constraint in LiDAR point cloud analytics and understanding.

Data augmentation (DA) [37, 3], which aims to expand the distribution of the training data by modifying and creating new training samples, has been widely studied for 2D images and demonstrated great potential in training robust DNN models with limited training images. However, most existing DA methods do not work well for LiDAR point clouds, a very meaningful but largely neglected task. Specifically, most existing DA methods perform *global augmentation* such as randomly scaling, flipping, and rotation which cannot augment local structures or model relationships across neighbouring

---

\*Shijian Lu is the corresponding author.

36th Conference on Neural Information Processing Systems (NeurIPS 2022).

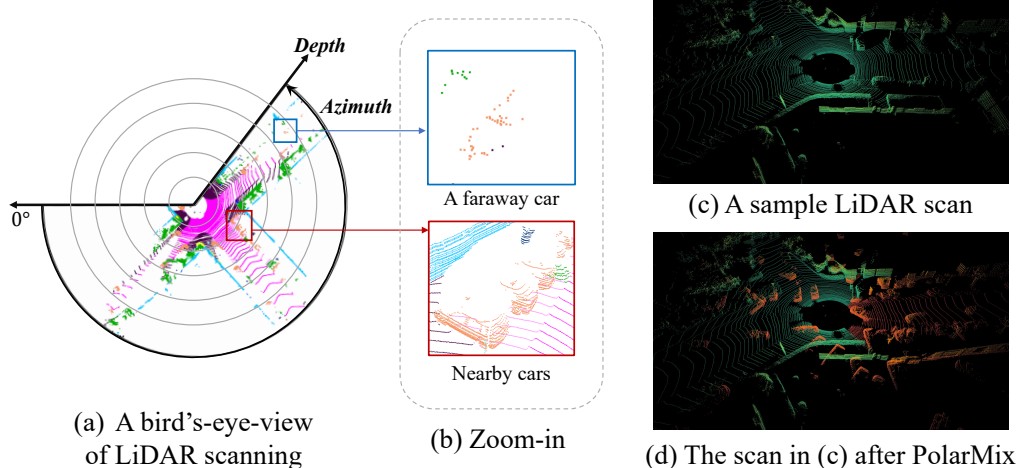

(a) A bird's-eye-view of LiDAR scanning

(b) Zoom-in

(c) A sample LiDAR scan

(d) The scan in (c) after PolarMix

Figure 1: A LiDAR sensor rotates and scans environments by the azimuth in XY plane, and the captured points (as shown in a bird's eye view in (a)) bear LiDAR-specific properties including partial visibility (i.e., only object sides facing the LiDAR sensor have points captured) and density variation along the depth as illustrated in close-up views in (b). PolarMix mixes points across LiDAR scans along the scanning direction which enriches point distribution while preserving data fidelity effectively. For a sample LiDAR scan in (c), (d) shows one of its augmentations with PolarMix where points in orange color are cropped and copies from another LiDAR scan.

point cloud scans. Recently, several studies [50, 49, 13] explore *local augmentation* that creates new training samples via cut and mix of 2D images. However, the *local augmentation* does not work well for point clouds as it does not consider the unique scanning mechanism of LiDAR sensors (e.g., via continuous 360-degree sweeping) and specific properties of the captured point data.

This work focuses on effective and efficient data augmentation for better learning from limited LiDAR point cloud data. To this end, we design *PolarMix*, a simple yet generic DA technique that can effectively work across different perception tasks and datasets. PolarMix achieves these unique features by capturing the essential properties of LiDAR point clouds, namely, partial visibility and density variation that are closely associated with the sweeping mechanism of LiDAR sensors. Specifically, objects in LiDAR scans are incomplete where only object sides facing the LiDAR sensor are scanned with points as illustrated in Fig. 1(a). In addition, point density varies with point depth as illustrated in Fig. 1(b). Effective data augmentation needs to cater for these LiDAR-specific features to ensure the fidelity and usefulness of the augmented point clouds in network training.

Inspired by the above observations, PolarMix crops, edits, and mixes points along the LiDAR scanning direction (*i.e.*, the *azimuth* in the 3D polar system in Fig. 1(a)) to enrich point cloud distributions while maintaining its fidelity. It consists of two cross-scan augmentation approaches. The first is scene-level swapping which exchanges point cloud sectors of two circular LiDAR scans that are cut along the azimuth axis as illustrated in Fig. 2(a). The second is instance-level rotation and paste which cuts point cloud instances from one LiDAR scan, rotates them along the scanning direction multiple times (to create multiple copies), and pastes the rotated instances into another LiDAR scan as illustrated in Fig. 2(b). For the sample LiDAR scan in Fig. 1(c), Fig. 1 (d) shows one of its augmentation where point cloud sectors and instances are mixed in with high fidelity. Extensive evaluations show that the PolarMix augmented point clouds improve the network training consistently across various tasks and benchmarks, more details to be described in experiment part.

The contribution of this paper can be summarized in three aspects. *First*, we introduce PolarMix, a simple yet effective point cloud augmentation technique that can enrich point cloud distributions while maintaining point cloud fidelity concurrently. *Second*, PolarMix is generally applicable and can work for different network architectures, perception tasks (e.g., object detection, semantic segmentation, etc.), and datasets/domains with consistent performance gains. *Third*, PolarMix is easy to use and can be incorporated as a plug and play by most existing point cloud networks. It also works well for unsupervised domain adaptation with state-of-the-art performance.

## 2 Related Works

**Data augmentation for 2D images**. Data augmentation has been widely studied across different 2D computer vision tasks such as image classification [15, 40], object detection [32], and semantic segmentation [4]. It plays an important role in effective and efficient deep network training since collecting and annotating training images is often laborious and time-consuming. One typical DA approach is global augmentation that aims to learn certain transformation invariance in image recognition tasks [33, 13], *e.g.*, random cropping [22, 38, 39], random scaling[38, 39], random erasing [52], color jittering [39], etc. Another typical DA approach is local augmentation which performs various mix operations to generate new training data. For example, mixup [50, 43] generates new data via convex combinations of the input pixels/feature embeddings and the output labels. CutMix [49] pastes rectangular crops from other images instead of mixing the whole images. Recently, several studies [51, 5, 10, 13] introduce the object concept into the cut and mix operations: [51] extends mixup and cutmix into object detection; [5, 10, 13] cut instances from one image and paste them into another for training better instance segmentation networks.

**Data augmentation for 3D point clouds**. Data augmentation of point clouds has also attracted increasing attention in recent years. Similar to 2D computer vision tasks, one direct approach is to adopt global augmentation in 3D space such as random scaling, rotation, and translation, which can be directly incorporated for expanding 3D objects [45, 2], indoor point clouds [8], and outdoor point clouds [1, 12, 30]. Several studies also explored to augment local structures of point clouds: PointAugment [25] introduces an auto-augmentation network for shape-wise transformation and point-wise displacement; PatchAugment [36] exploits data augmentation in local areas; PointWOLF [21] applies weighted transformations in local neighbourhoods for enhancing the diversity of 3D objects. However, the aforementioned works focus on object-level augmentation which are not suitable for scene-level point clouds such as those in autonomous driving.

Recently, several studies explored the idea of mixing for augmenting point clouds. For example, PointMixUp [6] interpolates 3D objects to create new samples for training. PointCutMix [24] replaces subsets of point objects with that of other objects to enrich training data. However, both work focuses on object-level augmentation only. Several studies [48, 11, 46, 29] also explore scene-level mix but they are constrained with specific vision tasks. For example, GT-Aug [48, 11] cuts instances and pastes them into other LiDAR scans for the object detection task (requiring 3D bounding boxes for object cutting); Mix3D [29] concatenates points of two scenes as an out-of-context augmentation for the specific task of semantic segmentation. As a comparison, the proposed PolarMix can perform both object-level and scene-level augmentation. More importantly, its designs are perfectly aligned with LiDAR-specific data properties including partial visibility and density variation with depth which guarantees superior fidelity and effectiveness of the augmented point clouds. Furthermore, PolarMix is generic and applicable to various computer vision tasks such as semantic segmentation and object detection, more details to be discussed in the experiment part.

## 3 PolarMix

**Problem statement**. Let $s \in \mathbb{R}^{N \times 4}$ and $y$ denote a LiDAR scan with $N$ points and its labels, respectively. Each point $p_i$ in $s$ is a $1 \times 4$ vector with a 3D Cartesian coordinate relative to the scanner $(x_i, y_i, z_i)$ and an intensity value of returning laser beam. The goal of PolarMix is to generate new training samples $(\tilde{s}, \tilde{y})$ by cutting and mixing across two training samples $(s^A, y^A)$ and $(s^B, y^B)$. The generated training samples $(\tilde{s}, \tilde{y})$ is used for network training with the original loss function.

**The polar coordinates.** We adopt a 3D polar coordinate system where the position of a point is defined by three numbers $(\theta, r, \phi)$: $\theta$ is the *azimuth* angle from x-axis to y-axis which defines the rotation scanning angle of LiDAR sweeping; $r$ denotes the *depth* which is the distance of the point to the LiDAR sensor; $\phi$ is the *inclination* angle between the z-axis and the point vector $(x_i, y_i, z_i)$.

**PolarMix for LiDAR data augmentation**. We designed two point cloud augmentation approaches in PolarMix including a scene-level swapping approach $Sw()$ and an instance-level rotate-paste approach $Rp()$ for mixing LiDAR scans $s^A, s^B$ and their labels $y^A, y^B$. The combination of the two augmentation approaches in PolarMix can be defined as

$$\begin{aligned} \tilde{s} &= Sw(s^A, s^B | \alpha, \beta) \oplus Rp(s^A, s^B | C, \Omega) \\ \tilde{y} &= Sw(y^A, y^B | \alpha, \beta) \oplus Rp(y^A, y^B | C, \Omega) \end{aligned} \tag{1}$$

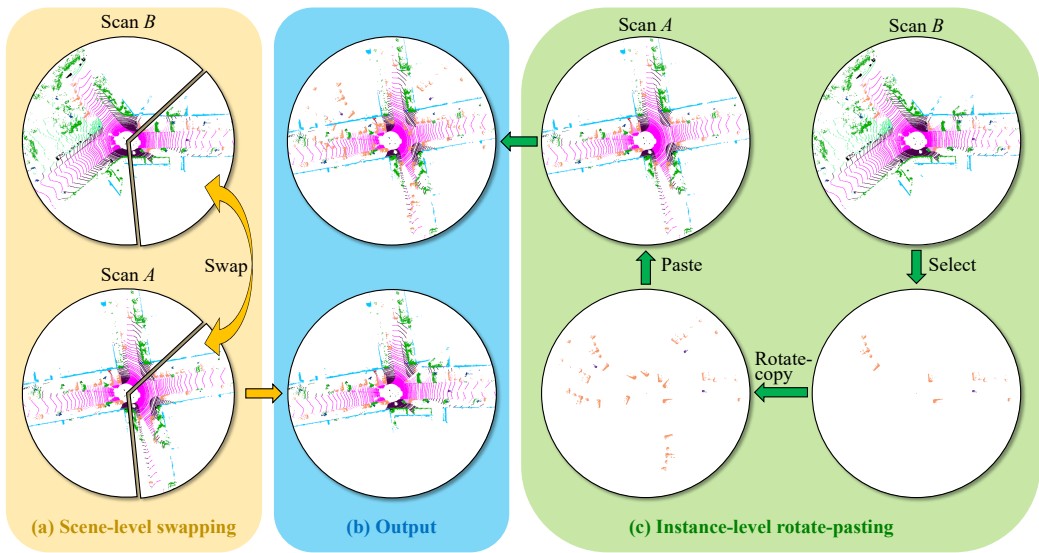

Scan *B*

Scan *A*

Swap

**(a) Scene-level swapping**

**(b) Output**

Scan *A*

Scan *B*

Paste

Select

Rotate-copy

**(c) Instance-level rotate-pasting**

Figure 2: The proposed *PolarMix* consists of two data augmentation designs: (a) The scene-level swapping exchanges sectors of LiDAR scans $A$ and $B$ that are cut with certain azimuth angles; (c) The instance-level augmentation cuts point instances from scan $B$, rotates them about the z-axis by multiple azimuth angles (for creating multiple copies of the cut point instances), and pastes the cut and rotated instances into scan $A$; The augmentations of scan $A$ by the two proposed augmentation approaches are shown in (b).

where $\oplus$ denotes a concatenation operation. $C$ and $\Omega$ represent class list and angle list for instance-level rotation and paste, respectively.

The scene-level swapping $Sw(s^A, s^B | \alpha, \beta)$ aims to cut a point cloud sector from azimuth angle $\alpha$ to azimuth angle $\beta$ from a LiDAR scan $s^A$, and switch it with the similarly cut point cloud sector from another LiDAR scan $s^B$. The swapping operation can be defined as follows

$$Sw(s^A, s^B | \alpha, \beta) = ((1 - \mathbf{M}_A^{\alpha,\beta}) \odot s^A) \oplus (\mathbf{M}_B^{\alpha,\beta} \odot s^B)$$
$$Sw(y^A, y^B | \alpha, \beta) = ((1 - \mathbf{M}_A^{\alpha,\beta}) \odot y^A) \oplus (\mathbf{M}_B^{\alpha,\beta} \odot y^B)$$

(2)

where $\mathbf{M}_A^{\alpha,\beta}$ denotes a binary mask indicating the azimuth range $[\alpha, \beta]$ that is cut out from LiDAR scan $s^A$, and $\odot$ is element-wise multiplication. The scene-level swapping thus exchanges two point cloud *sectors* of the same size from two LiDAR scans in a bird's-eye-view as illustrated in Fig. 2 (a) since the maximum scanning area in the horizontal plane is a circle. Note the angles $\alpha, \beta$ should be within the horizontal field-of-view of LiDAR sensor (*e.g.* within $360°$).

The instance-level augmentation $Rp(s^A, s^B | C, \Omega)$, as illustrated in Fig. 2 (c), crops point instances of semantic classes $C$ from LiDAR scan $s^B$, rotates them about z-axis by multiple azimuth angles $\Omega = \{\omega_1, \ldots, \omega_J\}$ to create multiple copies, and pastes all cropped and rotated point instances into another scan $s^A$. This augmentation operation can be defined by

$$Rp(s^A, s^B | C, \Omega) = s^A \oplus \sum_{\omega \in \Omega} \mathcal{R}_\omega(\mathbf{M}_B^C \odot s^B)$$
$$Rp(y^A, y^B | C, \Omega) = y^A \oplus \sum_{\omega \in \Omega} (\mathbf{M}_B^C \odot y^B)$$

(3)

where $\mathbf{M}_B^C$ is a binary mask indicating which semantic classes of points instances to crop from LiDAR scan $B$, and $\mathcal{R}_\omega$ represents the rotation matrix around the z-axis for an azimuth angle $\omega$.

Using a polar coordinate system to augment LiDAR points has two desirable features. First, it allows point cutting, rotating, and mixing to be perfectly aligned with the LiDAR scanning mechanism which greatly helps to preserve LiDAR-specific data properties such as partial visibility and density variation along the depth. Second, it simplifies the augmentation process with negligible computational overhead: The scene-level swapping involves point slicing and concatenating only while the

---

**Algorithm 1** PolarMix.

---

**Input:** Points and labels of two LiDAR scans: $\{s^A, y^A\}, \{s^B, y^B\}$; Class list and angle list for instance-level rotate and paste: $C, \Omega$; Azimuth range for scene-level swapping: $\alpha, \beta$.

**Output:** A new LiDAR scan for training: $\{\tilde{s}, \tilde{y}\}$.

1: $\tilde{s}, \tilde{y} = s^A, y^A$      *# Initialization*
2: **if** rand() $\leq \delta_1$ **then**      *# Scene-level swapping*
3:      Calculate azimuth $\theta$ for points in $\tilde{s}, s^B$ as $\tilde{\theta}, \theta^B$
4:      Delete points with labels in $\tilde{s}, \tilde{y}$ if $\alpha \leq \tilde{\theta} \leq \beta$
5:      Cut points with labels in $s^B, y^B$ if $\alpha \leq \theta^{\overline{B}} \leq \beta$
6:      Update $\tilde{s}, \tilde{y}$ by concatenating with cut points and labels from Scan $B$
7: **end if**
8: **if** rand() $\leq \delta_2$ **then**      *# Instance-level rotate-pasting*
9:      Copy points with labels from $s^B, y^B$ according to $C$
10:      **for** $\omega_j$ in $\Omega$ **do**
11:          Rotate copied points with $\mathcal{R}_{\omega_j}$, duplicate their labels
12:          Update $\tilde{s}, \tilde{y}$ by concatenating rotated points and labels
13:      **end for**
14: **end if**

---

instance-level augmentation can be achieved by simple dot products followed by point concatenation. Beyond that, PolarMix works in the input space which is naturally compatible with different network architectures. Algorithm 1 summarizes the pipeline of the proposed PolarMix.

**PolarMix for unsupervised domain adaptation**. The proposed PolarMix can be directly applied for unsupervised domain adaptation via self-training [54]. With LiDAR data from a labeled *source domain*, a supervised network model can be trained and applied to predict pseudo labels for LiDAR data from an unlabeled *target domain*. PolarMix can then cut and mix LiDAR scans between the source domain (with ground-truth labels) and the target domain (with pseudo labels). Such augmented LiDAR data mitigates the inter-domain discrepancy which facilitates unsupervised domain adaptation effectively, more details to be described in the ensuing experiments.

## 4 Experiments

We evaluate how PolarMix benefits deep neural network training for LiDAR point cloud understanding. In Section 4.1, we evaluate it over the semantic segmentation task across different deep architectures and benchmarking datasets. In Section 4.2, we evaluate it over object detection, mainly to examine its generalization capability across different computer vision tasks. In Section 4.3, we evaluate how it facilitates unsupervised domain adaptation over multiple synthetic-to-real domain adaptation benchmarks of LiDAR data. Finally, we provide an in-depth analysis of different components in PolarMix in Section 4.4.

### 4.1 PolarMix helps learn better representations for semantic segmentation

We first study how PolarMix helps learn better representation for semantic segmentation. The experiments were conducted over multiple deep architectures and public datasets.

#### 4.1.1 Experimental Settings

**Dataset.** We evaluate PolarMix over three LiDAR datasets of driving scenes that have been widely adopted for benchmarking in semantic segmentation. The first is **SemanticKITTI** [1] which is a large-scale dataset collected in a city of Germany. It has 43,551 LiDAR scans with 64 beams with point-wise annotations of 19 semantic classes. We follow the widely-adopted split and use sequences 00-07, 09-10 as the training set and sequence 08 for validation. The second is **nuScenes-lidarseg** [12] dataset which has 40,000 scans captured in 1000 scenes of 20s duration. It is collected with a 32 beams LiDAR sensor at 20Hz frequency with point-wise annotations of 16 semantic classes. We follow the officially split of training data and validation data. The third is **SemanticPOSS** [30] which consists of 2,988 annotated point cloud scans of 14 semantic classes. We follow the official benchmark setting,

Table 1: Semantic segmentation over the validation set of the dataset SemanticKITTI. The baseline with either MinkNet or SPVCNN does not involve any data augmentation. CGA means conventional global augmentation which includes random scaling and random rotation. The symbol † mean that the related local data augmentation is on top of CGA, e.g., +*CutMix*† means that the network training involves both CGA and CutMix. PolarMix achieves clearly the best semantic segmentation across both deep networks.

| Methods | car | bi.cle | mt.cle | truck | oth-v. | pers. | bi.clst | mt.clst | road | parki. | sidew. | oth-g. | build. | fence | veget. | trunk | terra. | pole | traf. | mIoU |
|---|---|---|---|---|---|---|---|---|---|---|---|---|---|---|---|---|---|---|---|---|
| MinkNet [7] | 95.9 | 3.7 | 44.9 | 53.2 | 42.1 | 53.7 | 68.9 | 0.0 | 92.8 | 43.0 | 80.0 | 1.8 | 90.5 | 60.0 | 87.4 | 64.5 | 73.3 | 62.1 | 43.7 | 55.9 |
| +CGA | 96.3 | 8.7 | 52.3 | 63.2 | 51.6 | 63.5 | 74.4 | 0.1 | 93.3 | 46.6 | 80.4 | 0.8 | 90.3 | 60.0 | 88.0 | 65.1 | 74.5 | 62.8 | 46.8 | 58.9(+3.0) |
| +CutMix† [49] | 96.0 | 10.2 | 59.3 | **78.7** | 52.1 | 63.4 | 79.4 | 0.0 | 93.5 | 47.8 | 80.7 | **1.6** | 90.3 | 61.0 | 87.5 | 66.2 | 73.3 | 64.0 | 46.8 | 60.6(+5.7) |
| +CopyPaste† [13] | **96.6** | 18.4 | 62.8 | 76.3 | **64.6** | 68.9 | 82.8 | 1.0 | 93.1 | 45.3 | 80.2 | 1.4 | 90.5 | 60.7 | 88.1 | 67.8 | 74.6 | 63.7 | 49.1 | 62.4(+6.5) |
| +Mix3D† [29] | 96.3 | 29.6 | 61.8 | 68.5 | 55.4 | **72.7** | 77.7 | 1.0 | **94.3** | **52.9** | **81.7** | 0.9 | 89.1 | 55.5 | 88.3 | **69.3** | 74.6 | **65.2** | **50.3** | 62.4(+6.5) |
| +PolarMix†(ours) | 96.3 | **51.2** | **75.6** | 63.4 | 63.9 | 71.9 | **85.6** | **4.9** | 93.6 | 45.8 | 81.4 | 1.4 | **91.0** | **62.8** | **88.4** | 68.5 | **75.0** | 64.6 | 49.9 | **65.0(+9.1)** |
| SPVCNN [41] | 94.9 | 9.1 | 55.8 | 66.5 | 33.7 | 61.8 | 75.9 | 0.2 | 93.1 | 45.3 | 79.6 | 0.4 | 91.4 | 62.7 | 87.5 | 66.2 | 72.9 | 62.8 | 42.7 | 58.0 |
| +CGA | 96.1 | 21.8 | 57.8 | **69.2** | 49.8 | 66.7 | 80.8 | 0.0 | 93.4 | 44.8 | 80.1 | 0.2 | 90.9 | 62.9 | 88.5 | 64.8 | 75.7 | 63.6 | 46.2 | 60.7(+2.7) |
| +CutMix† [49] | 96.1 | 21.4 | 59.6 | 71.2 | 54.2 | 66.8 | 81.8 | 0.0 | 93.5 | **49.6** | 81.1 | 2.2 | 90.9 | 63.1 | 87.9 | 66.9 | 74.1 | 63.8 | 49.8 | 61.7(+3.7) |
| +CopyPaste† [13] | 96.0 | 32.4 | 66.4 | 67.1 | 52.9 | 74.8 | 84.3 | 3.6 | 93.3 | 46.9 | 80.2 | **2.5** | 91.1 | **64.1** | 88.1 | 67.0 | 73.9 | 64.0 | **51.6** | 63.2(+5.2) |
| +Mix3D† [29] | **96.5** | 35.9 | 65.0 | 66.6 | 60.2 | 75.3 | 83.3 | 0.0 | **93.8** | 49.0 | 81.1 | 1.4 | 90.6 | 60.0 | **89.2** | **70.2** | **76.4** | **64.8** | 50.5 | 63.7(+5.7) |
| +PolarMix†(ours) | **96.5** | **53.9** | **79.7** | 68.5 | **64.9** | **75.6** | **87.8** | **7.5** | 93.5 | 47.3 | **81.2** | 1.1 | **91.2** | 63.8 | 88.2 | 68.2 | 74.2 | 64.5 | 49.4 | **66.2(+8.5)** |

*i.e.* sequence 03 for validation and the rest for training. For all semantic segmentation experiments, we adopt mean intersection-over-union (mIoU) as the evaluation metric.

**Architectures and implementation details.** We evaluate PolarMix over four widely adopted semantic segmentation networks: 1) MinkNet [7] which is a typical voxel-based sparse CNN; 2) SPVCNN [41] which is a hybrid network with a sparse convolutional and a point-based sub-network; 3) RandLA-Net [16] which is a standard point-based network; and 4) Cylinder3D [53] which a state-of-the-art cylindrical and asymmetrical 3D CNN. We adopt the default training hyper-parameters in the open-source repositories[234] for all four networks, and the only modification is the batch size for SPVCNN and MinkNet (we change it to 8). We conducted experiments with a single Tesla 2080Ti GPU for MinkNet and SPVCNN and a Tesla V100 GPU for RandLA-Net and Cylinder3D. Note training RandLA-Net and Cylinder3D takes relatively longer time, we therefore uniformly sub-sampled the same 10% of SemanticKITTI for faster experiments with these two networks.

For augmentation with scene-level swapping, we randomly crop 180° sectors from 360° for $[\alpha, \beta]$ for point swapping. For augmentation with instance-level rotate and paste, we take three rotation angles for dataset SemanticKITTI (0°, and another two rotation angles randomly picked from $(0°, 120°]$ and $(120°, 240°]$), and two rotation angles for datasets SemanticPOSS and nuScenes-lidarseg ($0°$ and another rotation angle randomly chosen from either $+90°$ or $-90°$). We set $\delta_1, \delta_2$ as 0.5, 1, respectively. We also examine hyper-parameters of PolarMix with details provided in the appendix.

### 4.1.2 Results

**PolarMix improves semantic segmentation by large margins**. Since data augmentation for LiDAR semantic segmentation is a relatively under-explored task with few existing works, we selected the highly-related mixing-based methods including Cut-Mix [49] and Copy-paste [13] in 2D vision and the pioneering work Mix3D [29] in 3D vision as baseline augmentation methods, and compared the proposed PolarMix with them. Tables 1 and 2 show experimental results across two networks *MinkNet* and *SPVCNN* and three datasets SemanticKITTI, nuScenes-lidarseg, and SemanticPOSS. It can be observed that the baseline with either MinkNet or SPVCNN (without involving any data augmentation in network training) produces fair semantic segmentation for all three datasets. However, including global augmentation with random scaling and rotation (i.e., +*CGA* in the two tables) improves semantic segmentation consistently across the two networks and the three datasets. On top of the global augmentation, further including local augmentation (i.e., +*CutMix*†, +*CopyPaste*†, and

---

[2]MinkNet and SPVCNN: https://github.com/mit-han-lab/spvnas
[3]RandLA-Net: https://github.com/QingyongHu/RandLA-Net
[4]Cylinder3D: https://github.com/xinge008/Cylinder3D

Table 2: Semantic segmentation over the validation set of the datasets nuScenes-lidarseg and SemanticPOSS. The baseline with either MinkNet or SPVCNN does not involve any data augmentation. CGA means conventional global augmentation which includes random scaling and random rotation. The symbol † mean that the related local data augmentation is on top of CGA, e.g., +CutMix† means that the network training involves both CGA and CutMix. PolarMix achieves clearly the best semantic segmentation across both deep networks.

| DA methods | MinkNet [7] | | SPVCNN [41] | |
| --- | --- | --- | --- | --- |
| | nuScenes-lidarseg | SemanticPOSS | nuScenes-lidarseg | SemanticPOSS |
| None | 67.1 | 52.1 | 68.4 | 50.7 |
| +CGA | 70.2(+3.1) | 55.1(+3.0) | 69.1(+0.7) | 55.3(+4.6) |
| +CutMix† [49] | 70.4(+3.3) | 56.0(+3.9) | 71.7(+3.3) | 54.7(+4.0) |
| +Copy-Paste† [13] | 70.8(+3.7) | 55.9(+2.8) | 71.3(+2.9) | 56.2(+5.5) |
| +Mix3D† [29] | 70.1(+3.0) | 55.3(+3.2) | 70.5(+2.1) | 54.4(+3.7) |
| +PolarMix†(Ours) | **72.0(+4.9)** | **57.4(+5.3)** | **72.1(+3.7)** | **58.6(+7.9)** |

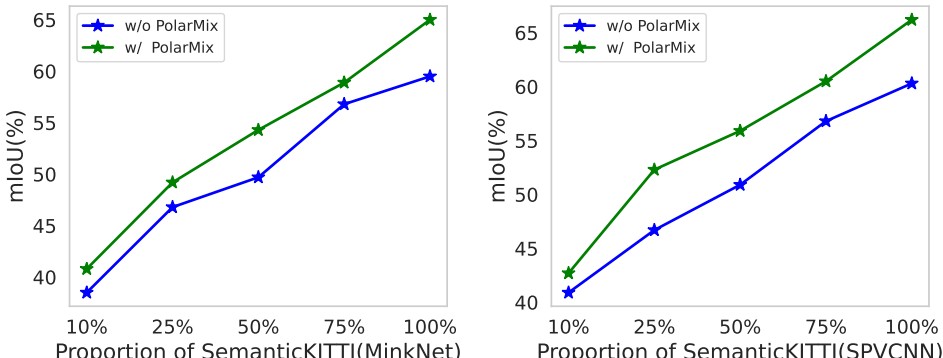

Figure 3: PolarMix helps reduce annotated training data effectively. For both MinkNet and SPVCNN, including PolarMix achieves similar segmentation accuracy by using around 75% annotated training data only, hence helps save around 25% efforts in training data collection and annotation.

+*Mix3D*†) further improves the semantic segmentation in most cases. As a comparison, PolarMix introduces the best performance gains consistently across the two baseline networks and the three evaluated benchmarking datasets, demonstrating its great robustness and generalization capabilities across different network architectures and datasets.

**PolarMix improves data-efficiency**. The proposed PolarMix works reliably with different amounts of training data, and it also improves data efficiency by reducing training data and annotations effectively. As shown in Fig. 3, the data augmentation with PolarMix consistently helps across different proportions of training data of SemanticKITTI as well as two different segmentation networks MinkNet and SPVCNN. In addition, including PolarMix can achieve similar segmentation mIoU while using 75% of SemanticKITTI in network training (as compared with training with 100% SemanticKITTI without using PolarMix) for both networks, hence saves 25% efforts in training data collection and annotations.

**PolarMix works across deep architectures**. The proposed PolarMix can work across different deep architectures beyond the voxel-based MinkNet and the sparse convolution network SPVCNN. We evaluate this feature by experimenting with two new deep architectures including the point-based network RandLA-Net [16] and the more recent 3D cylindrical convolutional architecture of Cylinder3D [53]. We train the two networks with default settings as in the officially released repositories. As shown in Table 3, incorporating PolarMix improves the segmentation performance consistently by large margins for both strong baselines (trained with 10% of SemanticKITTI data). This further verifies that PolarMix has superior generalization capability across different 3D deep architectures.

Table 3: Semantic segmentation results over the validation set of the SemanticKITTI dataset. We subsample the same 10% of the dataset for training. PolarMix consistently works across different 3D deep architectures.

| Methods | mIoU |
|---|---|
| RandLA-Net [16] | 45.4 |
| RandLA-Net [16]+PolarMix | 50.5(+5.2) |
| Cylinder3D [53] | 60.6 |
| Cylinder3D [53]+PolarMix | 62.5(+1.9) |

## 4.2 PolarMix helps learn effective representations for object detection

Table 4: Object detection results on the validation set of nuScenes dataset. Incorporating PolarMix into the network training consistently improves the object detection across three different deep frameworks including PointPillar, Second, and CenterNet.

| Methods | Car | Truck | Bus | Trailer | CV | Ped | Motor | Bicycle | TC | Barrier | mAP | NDS |
|---|---|---|---|---|---|---|---|---|---|---|---|---|
| PointPillar [23] | 80.4 | 45.0 | 54.0 | 25.4 | 10.5 | 71.0 | 36.0 | 9.4 | 44.8 | 42.8 | 41.8 | 54.9 |
| +PolarMix | 80.9 | 50.1 | 59.2 | 33.7 | 13.6 | 69.3 | 37.0 | 6.4 | 44.7 | 42.0 | 43.7(+1.9) | 55.7(+0.8) |
| Second [48] | 80.8 | 49.8 | 60.5 | 27.3 | 14.4 | 78.0 | 41.8 | 20.8 | 61.0 | 53.4 | 48.8 | 58.6 |
| +PolarMix | 81.3 | 53.6 | 68.3 | 34.4 | 20.0 | 76.5 | 38.2 | 14.7 | 59.6 | 56.8 | 50.3(+1.5) | 60.0(+1.2) |
| CenterNet[9] | 81.0 | 51.6 | 62.3 | 27.9 | 14.9 | 79.5 | 56.0 | 41.2 | 59.0 | 54.9 | 52.8 | 59.6 |
| +PolarMix | 80.4 | 53.4 | 68.8 | 32.5 | 17.5 | 79.1 | 58.3 | 44.1 | 57.7 | 62.2 | 55.4(+2.6) | 61.1(+1.5) |

**Setup.** The proposed PolarMix works in the input space with independence of specific tasks. We verify this property by evaluating it over object detection, another classical 3D understanding task that aims to predict 3D bounding box and label for each interested object instance. We perform experiments with dataset nuScenes [12] and three classical deep networks including PointPillar [23], Second [48], and CenterNet[9]. For implementation, we adopted default training hyper-parameters and optimizer in the OpenPCDet repository and training with two Tesla 2080Ti GPUs (11GB). We used random flip along the X and Y axis, random rotation, and random scaling for basic data augmentation. For a fair comparison, PolarMix is directly implemented on top of the baseline with the same configurations. For evaluation metrics, we adopted the widely used mean Average Precision (mAP) and nuScenes detection score (NDS).

**Results.** Table 4 shows experimental results. It can be observed that incorporating PolarMix improves both mAP and NDS consistently across the three tested deep networks. This experiment shows that the proposed PolarMix has superior generalization capability across different computer vision tasks, largely due to its cut-edit-mix strategy which enriches the distribution of the training data in the input space without changing LiDAR-specific data properties.

## 4.3 PolarMix helps reduce domain gap

Unsupervised domain adaptation (UDA) is an important research topic, aiming to solve the noticeable performance drops of deep neural networks while training and testing across different domains, as a result of the distribution bias (domain shift). UDA has been widely studied in both 2D vision [55, 56, 17, 18, 14, 19] and 3D vision [31, 20, 28, 46, 34, 35]. The proposed PolarMix can be easily extended for UDA by mixing labelled source point data and unlabeled target point data. We evaluate this nice feature by conducting experiments over two challenging synthetic-to-real point cloud segmentation benchmarks including SynLiDAR → SemanticKITTI and SynLiDAR → SemanticPOSS. SynLiDAR [46] is a synthetic LiDAR point cloud dataset that consists of 198k scans as collected from virtual scenes. It shares 19 common point classes with the SemanticKITTI and 13 common point classes with the SemanticPOSS. In the experiments, we train networks with the labelled SynLiDAR data (as the source data) and the unlabeled SemanticKITTI and SemanticPOSS data (as the target data), and perform evaluations over the validation set of SemanticKITTI and

Table 5: Experiments on unsupervised domain adaptation with SynLiDAR (as source) and SemanticKITTI and SemanticPOSS (as target). PolarMix achieves clearly the best semantic segmentation across both unsupervised domain adaptation setups.

| Methods | SynLiDAR → SemanticKITTI | SynLiDAR → SemanticPOSS |
|---|---|---|
| Source Only | 20.4 | 20.1 |
| ADDA [42] | 22.8 | 24.9 |
| Ent-Min [44] | 25.5 | 25.5 |
| Self-training [56] | 26.5 | 27.1 |
| PCT [46] | 28.9 | 29.6 |
| PolarMix(Ours) | **31.0** | **30.4** |

SemanticPOSS. We follow the existing benchmarks [46] and adopt MinkNet as the segmentation model.

We adopt the self-training approach as described in section 3 for unsupervised domain adaptation. Specifically, we first train a supervised model with the labelled source data and apply the supervised model to predict pseudo labels for the unlabelled target data. We then apply PolarMix to cut and mix between the labelled source data and the pseudo-labelled target data, and further train the model with all augmented point data. As shown in Table 5, incorporating PolarMix achieves state-of-the-art mIoUs for both SemanticKITTI and SemanticPOSS. The superior segmentation performance is largely attributed to the cut-and-mix strategy in PolarMix which effectively mitigates the distribution discrepancy across LiDAR scans of two different domains.

## 4.4 Ablation study

Table 6: Ablation study of PolarMix for semantic segmentation over SemanticKITTI dataset. SPVCNN is trained on the sequence 00 and and tested on the validation set.

| Methods | mIoU |
|---|---|
| SPVCNN [41] (baseline) | 48.9 |
| w/ Scene-level swapping | 50.8(+1.9) |
| w/ Intance-level pasting (simple-pasting) | 50.9(+2.0) |
| w/ Intance-level pasting (rotate-pasting) | 53.2(+4.3) |
| w/ PolarMix (complete) | 54.8(+5.9) |

We perform several ablation studies to examine the contribution of the two data augmentation components in the proposed PolarMix. In the ablation studies, we train SPVCNN with the sequence 00 of SemanticKITTI and evaluate the trained models over the validation set of SemanticKITTI. We adopt the same training configurations as described in Section 4.1 and Table 6 shows experimental results. With the conventional global augmentation including random rotation and random scaling, the trained SPVCNN model achieves a mIoU of 48.9%. On top of that, including the proposed scene-level swapping alone improves the mIoU by 1.4%. In addition, including the basic version of the proposed instance-level cut-and-mix (i.e., without multiple rotations to create multiple copies of the cropped object instances) alone improves the mIoU by 2.0% while incorporating the full instance-level cut-and-mix improves the mIoU by 4.3%. Finally, incorporating both augmentation components (i.e., the full PolarMix) improves the mIoU by 5.9%, demonstrating the complementary property of the two approaches in point data augmentation.

## 4.5 Discussion

We conducted experiments to examine whether mixing more than two LiDAR scans further improves the segmentation performances. Specifically, we increased the mixed LiDAR scans and benchmarking them without using PolarMix. The experiments were conducted with SPVCNN that is trained with sequence 00 of SemanticKITTI. As Table 7 shows, mixing two scans produces clearly the best performance. We examined the mixed data and found that mixing more scans introduces more

| #Scans | no mixing (baseline) | 2 | 3 | 4 |
|---|---|---|---|---|
| mIoU | 48.9 | **54.8** | 52.2 | 51.3 |

Table 7: Varying number of mixed scans. 'no mixing' represents the vanilla training without augmentation of PolarMix.

hardly distinguishable objects. The experimental results are consistent with other mixing-based augmentation works [50, 49, 6, 29].

## 5   Conclusion

This paper presents PolarMix which is a data augmentation method for LiDAR point cloud learning. It produces new point cloud data by mixing LiDAR scans for training networks. There are two approaches that are designed based on LiDAR data properties in PolarMix. Specifically, the scene-level swapping exchanges points within the same range of azimuth angles while the instance-level rotating-pasting selects points of certain classes from one LiDAR scan and rotates for multiple copies before pasting into another scan. Extensive experiments show the superiority of our method in data augmentation for both semantic segmentation and object detection across a variety of deep frameworks and public datasets. We also extended PolarMix into unsupervised domain adaptation and achieved state-of-the-art performances in multiple synthetic-to-real LiDAR data segmentation benchmarks. We hope that the idea of PolarMix can encourage future research to provide deeper insights on data augmentation for deep point cloud learning.

## Acknowledgments and Disclosure of Funding

This study is supported under the RIE2020 Industry Alignment Fund – Industry Collaboration Projects (IAF-ICP) Funding Initiative, as well as cash and in-kind contribution from Singapore Telecommunications Limited (Singtel), through Singtel Cognitive and Artificial Intelligence Lab for Enterprises (SCALE@NTU).

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
