# PolarMix: A General Data Augmentation Technique for LiDAR Point Clouds (Supplemental Material)

## 1   Implementation details

As described in section 4 of the manuscript, we compare PolarMix with state-of-the-art mixing-based methods including *CutMix* [6], *Copy-Paste* [1], and *Mix3D* [2] for semantic segmentation of LiDAR point clouds. CutMix and Copy-Paste are methods from 2D vision, and we extended them into 3D vision for benchmarking. Specifically, in CutMix we randomly cut rectangles instead of sectors as in PolarMix; In Copy-Paste we copy instances from one scan and shifts along the x-axis or y-axis before pasting them into the other scan. Mix3D is a 3D method that directly concatenates two scenes for the mixing. It is an extension of mixup [7] in the 3D space. We first implement global augmentation approaches including random rotation and random scaling on two LiDAR scans separately and then concatenate them for training. We use the same training configurations as the baseline for fair comparison for all three methods.

## 2   Parameter learning

| Swapping angular range | mIoU |
|---|---|
| No swapping | 48.9 |
| Random $45°$ | 48.0 |
| Random $90°$ | 49.4 |
| Random $135°$ | 50.3 |
| Random $180°$ | 50.8 |

(a) **Scene-level swapping**.

| Rotate pasting times | mIoU |
|---|---|
| No pasting | 48.9 |
| $\times 1$ | 50.9 |
| $\times 2$ | 52.4 |
| $\times 3$ | 53.2 |
| $\times 4$ | 52.3 |

(b) **Instance-level rotate-pasting**.

Table 1: Parameter analysis in PolarMix.

We conduct experiments to examine the effects of using different parameters in PolarMix. We adopt the same configuration in ablation study (section 4.4 in the manuscript), *i.e.* we train SPVCNN with the same training hyper-parameters on sequence 00 of SemanticKITTI and evaluate over the validation set of SemanticKITTI.

We first study how the azimuth angular range in the scene-level swapping approach ($\beta - \alpha$) affects augmentation effects. Specifically, we randomly crop different angular range of sectors from $360°$ for point swapping. Table 1 (a) lists results. We can see that randomly swapping sectors of $45°$ slightly downgraded segmentation performances from 48.9% mIoU to 48.0%. With the increase of the angular range, performances gains increased and swapping $180°$ reaches the best mIoU improvement (+1.9%).

We then study the augmentation effects when different numbers of rotate copies ($\Omega$) are pasted in the instance-level rotate-pasting approach. Table 1 (b) shows the results. We can see that pasting different numbers of copies of instances from other scans achieved significant mIoU gains continuously. The more copies the better segmentation performance as shown in '$\times 1, \times 2, \times 3$' in the table, which indicates the effectiveness of the approach in enriching data distribution. However, copying four times instances ($\times 4$ in the table) sees a slight drop in mIoU gain, largely due to the generated points being too dense to maintain data structural integrity.

36th Conference on Neural Information Processing Systems (NeurIPS 2022).

# 3 Broader impact

Better learning of LiDAR data will lead to safer autonomous vehicles. It contributes to earlier danger detection and prevention which can avoid traffic accidents. Besides, LiDAR data has also been widely applied in robotics and remote sensing, better semantic segmentation and 3D detection for LiDAR data can improve efficiency in economics and time in many cases, such as rescuing, survey, navigation, and so on.

# 4 Quantitative results

We provide quantitative results of semantic segmentation in SemanticKITTI. Fig. 1 and Fig. 2 show predictions of SPVCNN trained with or without PolarMix. We can see that PolarMix helps to achieve better segmentation aligned with the results in Table 1 in manuscript.

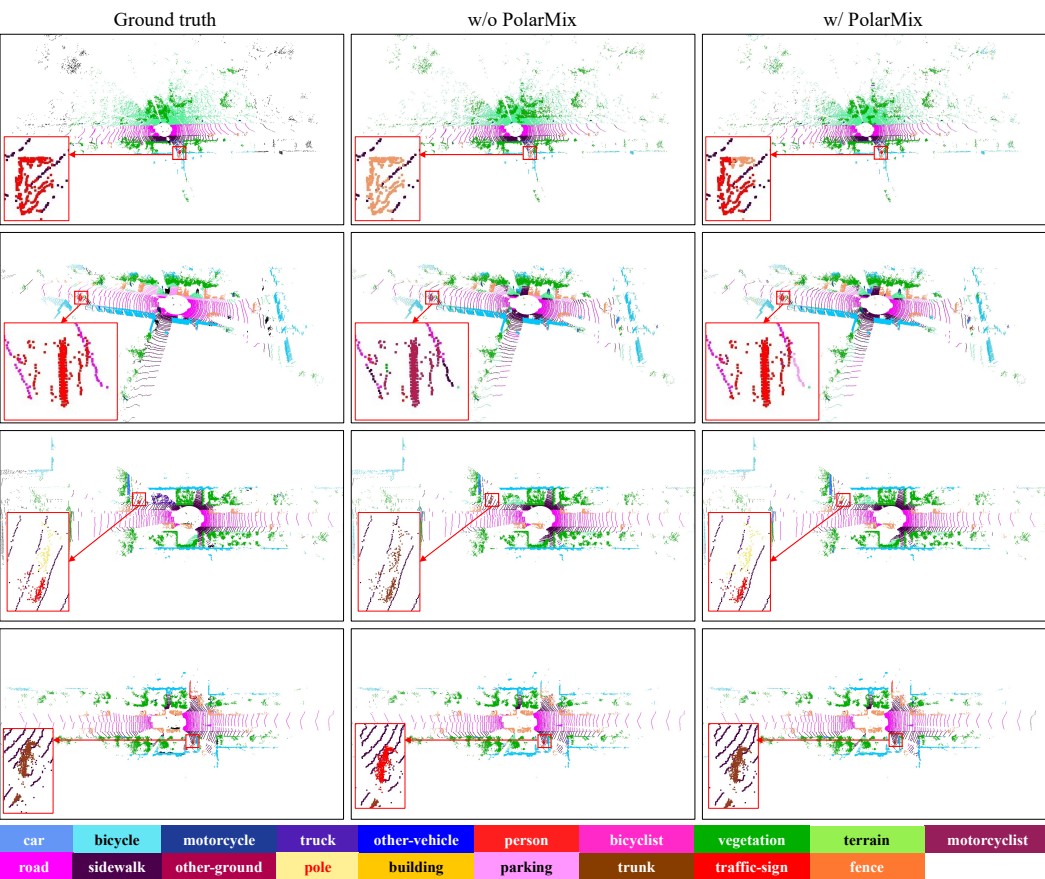

Figure 1: Illustration of semantic segmentation of SemanticKITTI point cloud by SPVCNN. The left column are examples with ground-truth segmentation; The middle column show predictions of SPVCNN; The right column show predictions of SPVCNN trained with our PolarMix. We zoom in areas in red boxes for better illustration. PolarMix can achieve better segmentation results.

# 5 Analysis

In this section, we conducted experiments to analyze how PolarMix benefits LiDAR point cloud learning.

PolarMix increases the recognition robustness in spatial locations: We randomly rotate instances in the testing LiDAR scans and report segmentation performances of MinkNet w/ or w/o using PolarMix,

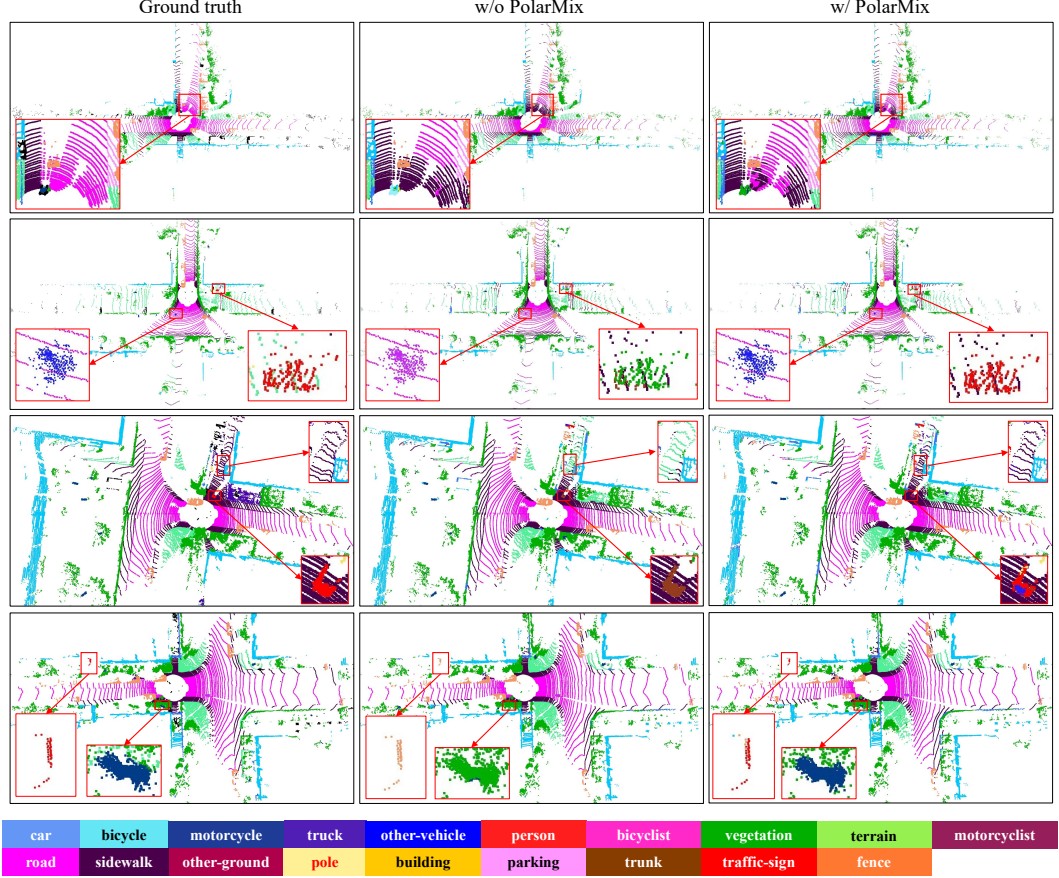

| Ground truth | w/o PolarMix | w/ PolarMix |
|---|---|---|

| car | bicycle | motorcycle | truck | other-vehicle | person | bicyclist | vegetation | terrain | motorcyclist |
|---|---|---|---|---|---|---|---|---|---|
| road | sidewalk | other-ground | pole | building | parking | trunk | traffic-sign | fence | |

Figure 2: Illustration of semantic segmentation of SemanticKITTI point cloud by SPVCNN. The left column are examples with ground-truth segmentation; The middle column show predictions of SPVCNN; The right column show predictions of SPVCNN trained with our PolarMix. We zoom in areas in red boxes for better illustration. PolarMix can achieve better segmentation results.

which evaluates how models recognize instances appearing in different spatial locations. Table 2 below shows experimental results. It can be seen that the baseline performance drops while rotating instances by different angles. This is largely because the baseline is very sensitive to the spatial location of instances that is often severely imbalanced in most existing datasets (due to LiDAR data collection and annotation constraints). As a comparison, PolarMix is more robust to the instance spatial location without much performance drop. The experimental results show that PolarMix effectively improves the generalization of the trained LiDAR model (with respect to the instance spatial location) by generating lots of training samples at different spatial locations.

| method | $0°$ | $45°$ | $90°$ | $135°$ | $180°$ |
|---|---|---|---|---|---|
| baseline | 58.9 | 58.0(-0.9) | 57.6(-1.3) | 57.9(-1.0) | 57.5 (-1.4) |
| +PolarMix | 65.0 | 64.9(-0.1) | 64.9(-0.1) | 65.0(-0.0) | 64.8(-0.2) |

Table 2: Segmentation with MinkUNet over the validation set of SemanticKITTI with rotated instances. PolarMix improves the robustness of the baseline clearly with respect to the angular variations of instances (i.e. spatial location variations).

PolarMix increases the recognition robustness in scene layout: We randomly swap sectors of two testing scans with different angles, which generates new testing LiDAR scans with different layouts of road scenes. Similarly, we report segmentation performances of MinkNet w/ or w/o using PolarMix. The results are summarized in Table 3 below. We can observe that the baseline performance drops significantly while swapping sectors in the testing set with different angular ranges. However, the

models trained with PolarMix are more robust with much less performance drop, indicating that PolarMix improves the robustness of LiDAR models with respect to the scene layout effectively.

| method | 0° | 45° | 90° | 135° | 180° |
|---|---|---|---|---|---|
| baseline | 58.9 | 58.0(-0.9) | 56.4(-2.5) | 56.6(-2.3) | 57.6(-1.3) |
| PolarMix | 65.0 | 64.5(-0.5) | 64.5(-0.5) | 64.2(-0.8) | 64.4(-0.6) |

Table 3: Segmentation result of MinkUNet over validation set of SemanticKITTI. We swap sectors of testing LiDAR scans to diversify layouts of road scenes and report mIoU performances. PolarMix significantly increases the robustness of the segmentation model.

PolarMix achieves better performance gains over closer points: We evaluated performance gains of PolarMix in recognizing points across different depth. Specifically, we split points of different ranges of depth and report segmentation performances of MinkNet over each split. The experimental results are shown in the Table 4. We can see that the performance gains decreases with the increase of depth.

| Depth (in meter) | [0,20) | [20, 40) | [40, 60) | [60, 80] |
|---|---|---|---|---|
| MinkUNet(baseline) | 61.0 | 48.6 | 29.6 | 47.7 |
| +PolarMix | 66.8(+5.8) | 54.8(+6.2) | 34.0(+4.4) | 48.8(+1.1) |

Table 4: Segmentation performances of MinkNet over points in different ranges of depth.

# 6 PolarMix for unsupervised domain adaptation

As illustrated in Section 4.3, we extend PolarMix for unsupervised domain adaptation. We adopted the typical self-training strategy which treats confident pseudo labels of target data as ground truth (i.e., the top 20% of the highest prediction scores initially) and then applies them to re-train the segmentation networks. The whole training repeats the two processes for five rounds (2 epochs in each round) with a gradually increasing confidence threshold (adding 5% after each round). The per-class segmentation result of Table 5 of the manuscript are shown in Table 5 and Table 6.

| Method | car | bi.cle | mt.cle | truck | oth-v. | pers. | bi.clst | mt.clst | road | parki. | sidew. | oth-g. | build. | fence | veget. | trunk | terra. | pole | traf. | mIoU |
|---|---|---|---|---|---|---|---|---|---|---|---|---|---|---|---|---|---|---|---|---|
| Source-Only | 42.0 | 5.0 | 4.8 | 0.4 | 2.5 | 12.4 | 43.3 | 1.8 | 48.7 | 4.5 | 31.0 | 0.0 | 18.6 | 11.5 | 60.2 | 30.0 | 48.3 | 19.3 | 3.0 | 20.4 |
| ADDA [3] | 52.5 | 4.5 | 11.9 | 0.3 | 3.9 | 9.4 | 27.9 | 0.5 | 52.8 | 4.9 | 27.4 | 0.0 | 61.0 | 17.0 | 57.4 | 34.5 | 42.9 | 23.2 | 4.5 | 22.8 |
| Ent-Min [4] | 58.3 | 5.1 | 14.3 | 0.6 | 1.8 | 14.3 | 44.5 | 0.5 | 50.4 | 4.3 | 34.8 | 0.0 | 48.3 | 19.7 | 67.5 | 34.8 | 52.0 | 33.0 | 6.1 | 25.5 |
| ST [8] | 62.0 | 5.0 | 12.4 | 1.3 | 9.2 | 16.7 | 44.2 | 0.4 | 53.0 | 2.5 | 28.4 | 0.0 | 57.1 | 18.7 | 69.8 | 35.0 | 48.7 | 32.5 | 6.9 | 26.5 |
| PCT [5] | 53.4 | 5.4 | 7.4 | 0.8 | 10.9 | 12.0 | 43.2 | 0.3 | 50.8 | 3.7 | 29.4 | 0.0 | 48.0 | 10.4 | 68.2 | 33.1 | 40.0 | 29.5 | 6.9 | 23.9 |
| PCT+ST [5] | 70.8 | 7.3 | 13.1 | 1.9 | 8.4 | 12.6 | 44.0 | 0.6 | 56.4 | 4.5 | 31.8 | 0.0 | 66.7 | 23.7 | 73.3 | 34.6 | 48.4 | 39.4 | 11.7 | 28.9 |
| PolarMix(ours) | 76.3 | 8.4 | 17.8 | 3.9 | 6.0 | 26.6 | 40.8 | 15.9 | 70.3 | 0.0 | 44.4 | 0.0 | 68.4 | 14.7 | 69.6 | 38.1 | 37.1 | 40.6 | 10.6 | 31.0 |

Table 5: Experiments on unsupervised domain adaptation with SynLiDAR (as source) and SemanticKITTI (as target).

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

| Method | pers. | rider | car | trunk | plants | traf. | pole | garb. | buil. | cone. | fence | bike | grou. | mIoU |
|---|---|---|---|---|---|---|---|---|---|---|---|---|---|---|
| Source-Only | 3.7 | 25.1 | 12.0 | 10.8 | 53.4 | 0.0 | 19.4 | 12.9 | 49.1 | 3.1 | 20.3 | 0.0 | 59.6 | 20.1 |
| ADDA [3] | 27.5 | 35.1 | 18.8 | 12.4 | 53.4 | 2.8 | 27.0 | 12.2 | 64.7 | 1.3 | 6.3 | 6.8 | 55.3 | 24.9 |
| Ent-Min [4] | 24.2 | 32.2 | 21.4 | 18.9 | 61.0 | 2.5 | 36.3 | 8.3 | 56.7 | 3.1 | 5.3 | 4.8 | 57.1 | 25.5 |
| ST [8] | 23.5 | 31.8 | 22.0 | 18.9 | 63.2 | 1.9 | 41.6 | 13.5 | 58.2 | 1.0 | 9.1 | 6.8 | 60.3 | 27.1 |
| PCT [5] | 13.0 | 35.4 | 13.7 | 10.2 | 53.1 | 1.4 | 23.8 | 12.7 | 52.9 | 0.8 | 13.7 | 1.1 | 66.2 | 22.9 |
| ST + PCT [5] | 28.9 | 34.8 | 27.8 | 18.6 | 63.7 | 4.9 | 41.0 | 16.6 | 64.1 | 1.6 | 12.1 | 6.6 | 63.9 | 29.6 |
| PolarMix (ours) | 32.6 | 39.1 | 25.0 | 11.9 | 64.2 | 5.8 | 29.6 | 15.3 | 44.8 | 13.3 | 23.8 | 10.7 | 79.0 | 30.4 |

Table 6: Experiments on unsupervised domain adaptation with SynLiDAR (as source) and Semantic-POSS (as target).

[3] Eric Tzeng, Judy Hoffman, Kate Saenko, and Trevor Darrell. Adversarial discriminative domain adaptation. In *Proceedings of the IEEE conference on computer vision and pattern recognition*, pages 7167–7176, 2017.

[4] Tuan-Hung Vu, Himalaya Jain, Maxime Bucher, Matthieu Cord, and Patrick Pérez. Advent: Adversarial entropy minimization for domain adaptation in semantic segmentation. In *Proceedings of the IEEE/CVF Conference on Computer Vision and Pattern Recognition*, pages 2517–2526, 2019.

[5] Aoran Xiao, Jiaxing Huang, Dayan Guan, Fangneng Zhan, and Shijian Lu. Transfer learning from synthetic to real lidar point cloud for semantic segmentation. *Proceedings of the AAAI Conference on Artificial Intelligence*, 36(3):2795–2803, Jun. 2022.

[6] Sangdoo Yun, Dongyoon Han, Seong Joon Oh, Sanghyuk Chun, Junsuk Choe, and Youngjoon Yoo. Cutmix: Regularization strategy to train strong classifiers with localizable features. In *Proceedings of the IEEE/CVF international conference on computer vision*, pages 6023–6032, 2019.

[7] Hongyi Zhang, Moustapha Cisse, Yann N Dauphin, and David Lopez-Paz. mixup: Beyond empirical risk minimization. *arXiv preprint arXiv:1710.09412*, 2017.

[8] Yang Zou, Zhiding Yu, Xiaofeng Liu, BVK Kumar, and Jinsong Wang. Confidence regularized self-training. In *Proceedings of the IEEE/CVF International Conference on Computer Vision*, pages 5982–5991, 2019.