# OpenReview forum: "PolarMix: A General Data Augmentation Technique for LiDAR Point Clouds"
_NeurIPS.cc/2022/Conference — NeurIPS 2022 Accept_

### Official Review · Reviewer_gjSm · 2022-07-10

**Rating:** 4
**Confidence:** 4
**Soundness:** 2 fair
**Presentation:** 2 fair
**Contribution:** 3 good

**Summary:**

This paper proposed a data augmentation method named PolarMix for LiDAR point cloud perception. It includes two separate operations: a scene-level swapping that first cuts point cloud sectors of two LiDAR scans w.r.t. the point azimuth values and then switches the cut sectors to form a new sample for training; and an instance-level copy-paste which selects instance points for certain classes from one scan, rotates them along the LiDAR scanning direction and pastes the rotated points to another scan. Experimental results show that PolarMix yields improvements for both LiDAR semantic segmentation and 3D object detection.

**Questions:**

Regarding weakness 1, there are some concerns w.r.t. the idea proposed in this work:
1. In L56, the authors state that PolarMix can maintain high fidelity for the augmented samples. However, the augmented scan shown in Figure 1 (d) contains lots of instances like cars that are scattering around the center-vehicle with arbitrary directions, which might not happen in real-world scenarios. Also, the pasted instances seem to be overlapped with the background classes. As instances like cars, bicycles, and pedestrians tend to appear only on the road surface, how to ensure that they are pasted to a proper position with high fidelity?

2. In L109, the LiDAR points in the polar coordinate system are defined by azimuth theta, depth r, and inclination phi. Why select only the azimuth to split point cloud sectors, but not depth or inclination or even the combination of all three?

3. In L207, the authors state that the proposed method can work across different LiDAR point cloud representations. Can it be applied to range view networks? How does it ensure a proper projection after concatenating or pasting new points in the current LiDAR point cloud?

4. In suppL19, randomly swapping sectors of 45 degrees could downgrade the segmentation performance. Any possible explanation on why this problem happens?

Regarding weakness 2, there are some concerns w.r.t. the experiment part of this work:
1. The main results for LiDAR semantic segmentation, e.g., Table 1, are from relatively old segmentation methods MinkNet (55.9% mIoU) and SPVCNN (58.0% mIoU), which do not yield competitive performance over state-of-the-art ones, such as Cylinder3D (65.9% mIoU). Although the authors involved RandLA-Net and Cylinder3D in later analysis, i.e., Table 3, they are only tested under the 10% data setting. Therefore, the performance gains of PolarMix from the relatively lower-score methods are not representative enough to demonstrate superiority.

2. The results for 3D object detection, i.e., Table 4, only include comparisons with the baselines. Other augmentation techniques, such as copy-paste, are recommended to include for a more comprehensive comparison.

3. The ablation studies in Section 4.4 is not comprehensive enough to support the effectiveness of each of the components in PolarMix. Besides, the scene-level swapping seems less effective compared to the rotate-pasting.

Regarding weakness 3, there are some minor problems w.r.t. the elaboration:
1. Omega and C should be properly defined after their first appearance, i.e., Eq. (1) in L116.

2. The delta_1 and delta_2 symbols in Algorithm 1 are not defined.

3. The sigma_1 and sigma_2 symbols in L185 are not defined.

4. In L247, why the cut-and-mix strategy in PolarMix can mitigate the domain discrepancy? Since PolarMix does not include any domain alignment components like the domain discriminator, the performance gains seem more related to the augmentation effect but not the adaptation effect.


**Limitations:**

This paper does not include a specific description of potential limitations. In L396, the author state that the effectiveness of the proposed method might be influenced by the parameter setting, which should be regarded as and included in ablation analysis. More general limitations are recommended to be discussed, such as the applicability of the proposed method for different types of LiDAR point clouds and networks.

**Strengths And Weaknesses:**

Strengths:
1. This paper introduces a new data augmentation method for LiDAR point clouds.

2. The proposed PolarMix is tested for both LiDAR semantic segmentation and 3D object detection and the results show that the proposed method can improve the baselines for both tasks.

Weaknesses:
1. This paper explains mostly the “what” and “how” but not much “why” for their specific augmentation operations.  More design insights and analysis can improve this paper a lot.

2. The experimental analysis is insufficient to support all conclusions stated in this paper.

3. The elaboration of this paper is not good enough. Some assumptions are made without clear logical relations.

---

> ### Author Response · Authors · 2022-08-02
> **Thank you for your confirmation of the value of our proposed method as well as the impressive experimental results.**
>
> Thank you for your confirmation of the value of our proposed method as well as the impressive experimental results. Below please find our responses regarding your concerns.
>
> W1-Q1: Why PolarMix maintain high fidelity of augmented LiDAR data \& our design insights?
>
> - We would clarify that the "high-fidelity" means that PolarMix allows augmenting LiDAR point clouds without impairing two typical LiDAR point properties (Lines 41-47) including:
> 	&emsp; 1: Objects in LiDAR scans are incomplete where only object sides facing the LiDAR sensor are scanned with points as illustrated in Fig. 1(a) in the submitted manuscript.
>     &emsp; 2: Point density decreases with the increase of depth as illustrated in Fig. 1(b) in the submitted manuscript.
> Previous data augmentation methods impair these two data properties severely which affects their effectiveness. E.g., CutMix[37] and Copy-paste[13] from 2D images work in the Cartesian coordinate system, Mix3D [21] mixes 3D scenes globally in an out-of-context manner. Extensive experiments in Section 4 of our submitted manuscript show that the PolarMix-augmented LiDAR data preserve the data fidelity and help train better segmentation models consistently.
> - In addition, multiple cars scattered around the center-vehicle may happen in real scenarios, e.g., in a traffic jam or accident. These situations are crucial for autonomous driving because they are closely related to dangerous situations. Please note that for data augmentation, we do not ensure that instances only appear in the "frequently" appeared locations. On the contrary, we enrich the diversity by allowing instances to appear at different locations. PolarMix generates new training samples with abundant instances in different locations thus enlarging the training distribution which enhances the robustness of recognition models.
> - We would clarify that some pasted instances overlapping with the background points is not a problem for point data augmentation. Recent studies in Mix3D [21] showed that such mixing would not downgrade but augment point-based models. Similar discontinuities can also be found in augmented images of CutMix[37] in 2D vision.
>
> ---
>
> W1-Q2: Mixing in depth and inclination directions?
>
> - Thank you for your suggestion! We also considered mixing in the depth and inclination directions in the stage of methodology design. After some preliminary studies, we found that mixing in these two dimensions either impairs LiDAR data fidelity or requires very complicated point rendering design to preserve LiDAR characteristics, e.g., for preserving partial visibility and density variation along the depth. As a comparison, cutting and mixing along the azimuth direction preserves point fidelity greatly and efficiently as it is well aligned with the point capturing process.
>
> ---
>
> W1-Q3: Does PolarMix work for range view networks?
>
> - We conducted the suggested experiment and evaluated PolarMix over SalsaNext [a], one of the most advanced projection-based methods. Since training SalsaNext with full SemanticKITTI training set takes more than one week, we report performances of 10\% of training data for fast experiments. The mIoU of SalsaNext w/o and w/ PolarMix are 52.2 and 54.7, respectively, indicating the effectiveness of PolarMix in augmenting range view networks.
> - Our PolarMix works in the input space and it is model-agnostic. The range projection of the PolarMix-augmented takes the same process for raw data, e.g. spherical projection for SalsaNext.
>
> [a] Cortinhal T, Tzelepis G, Erdal Aksoy E. SalsaNext: Fast, uncertainty-aware semantic segmentation of LiDAR point clouds.
>
> ---
>
> W1-Q4: Why randomly swapping sectors of 45 degrees downgrades?
>
> - Swapping small angular ranges such as $45^\circ$ may damage semantic layouts of LiDAR scenes, leading to the downgraded performances of segmentation models. The experiment in Table 1 of the supplementary material shows that wider ranges of sectors should be selected for swapping.

---

> > ### Author Response · Authors · 2022-08-02
> > **Continued from previous responses**
> >
> > W2-Q1: Is PolarMix a superior DA method?
> >
> > - Yes PolarMix is a superior DA method which is model-agnostic and achieves excellent point segmentation performance gains. As shown in the table below (extracted from Table 1 of the submitted manuscript), SPVCNN w/ PolarMix achieves a mIoU of 66.2\% mIoU over the validation set of SemanticKITTI dataset, which outperforms the state-of-the-art Cylinder3D [41] at 65.9\% as reported in Table 3 in [41]. The experiments show that PolarMix is indeed a superior and SOTA augmentation method for LiDAR point cloud learning.
> > - As commented by other reviewers, we tested over many baselines, datasets, and tasks to verify the generalization of PolarMix, which involved a huge amount of training resources. Since training Cylinder3D takes a very long training time (more than 2 weeks as in Lines 178-180), we only report the performance gain of PolarMix over Cynlinder3D with 10\% of training data, as shown in Table 3 in the submitted manuscript. We will report the performance of Cylinder3D+PolarMix with full training data in the updated manuscript.
> >
> > | Method               | mIoU      |
> > | :------------------- | :-------- |
> > | Cylinder3D[41]       | 65.9 (copied from [41])      |
> > | SPVCNN [30]  | 60.7 |
> > | SPVCNN+PolarMix | **66.2** |
> >
> > ---
> >
> > W2-Q2: Comparison with other augmentation techniques, such as copy-paste, in Table 4?
> >
> > - Thanks for your suggestion. We conducted the suggested experiment and include the performance of PointPillar with Copy-Paste over validation set of nuScenes dataset. As the results in the table below, PolarMix achieves a clearly performance gain and surpasses Copy-Paste, which is consistent with our experimental results in the segmentation task (Tables 1,2 in the submitted manuscript).
> >
> > | Methods | mAP | NDS |
> > | :------ | :---- | :---- |
> > | PointPillar [18] |  41.8 | 54.9 |
> > | +CopyPaste [13] (newly-included) | 42.9(+1.1) | 55.2(+0.3) |
> > | +PolarMix | **43.7(+1.9)** | **55.7(+0.8)** |
> >
> > ---
> >
> > W2-Q3: Is the ablation study supportive?
> >
> > - As shown in the table below (Table 6 in manuscript), both approaches in PolarMix improve segmentation performances of SPVCNN clearly: The scene-level swapping improves segmentation performance with 1.9\% and instance-level pasting improves 4.3\%. These two approaches are complementary and the complete PolarMix achieves the best results at 54.8\%(+5.9\%).
> >
> > | Methods | mIoU |
> > | :------ | :------ |
> > | SPVCNN (baseline) | 48.9 |
> > | w/ Scene-level swapping | 50.8(+1.9)|
> > | w/ Intance-level pasting (simple-pasting) | 50.9(+2.0) |
> > | w/ Intance-level pasting (rotate-pasting) | 53.2(+4.3) |
> > | w/ PolarMix (complete) | **54.8(+5.9)** |
> >
> > ---
> >
> > W3: Minor problems w.r.t. the elaboration?
> >
> > - Thanks for your meticulous effort in details. The $\sigma_1$ and $\sigma_2$ in L185 are typos and should be $\delta_1$ and $\delta_2$ in the algorithm 1. We will revise the manuscript for clearer representation.
> > - PolarMix mitigates the domain shift by creating new intermediate domains of point clouds from two different domains. It improves pseudo-label accuracy for self-training.

---

> ### Author Response · Authors · 2022-08-08
> **To Reviewer gjSm:**
>
> Dear Reviewer gjSm:
>
> We thank you for the precious review time and valuable comments. We have provided corresponding responses and results, which we believe have covered your concerns. We hope to further discuss with you whether or not your concerns have been addressed. Please let us know if you still have any unclear parts of our work.
>
> Best regards,
> Authors

---

### Official Review · Reviewer_YDbX · 2022-07-11

**Rating:** 6
**Confidence:** 5
**Soundness:** 3 good
**Presentation:** 3 good
**Contribution:** 2 fair

**Summary:**

- This paper introduces an approach to augmenting cylindrical LiDAR point cloud to acquire boosted performance on 3D semantic segmentation and 3D detection. The proposed approach called PolarMix enables cut, edit, and mix point clouds along the scanning direction. The augmentation happens on the scene-level and instance-level so that the augmented data provides a variety of combinations of the augmented scenes. The proposed approach is superior to conventional global rotation and scale augmentation, CutMix [37], Copy-Paste [13], and Mix3D [21].

**Questions:**

1. Did the authors try to mix more than two scenes? The proposed Algorithm only shows how to mix two scans, but I wonder if the idea could be generalized.
2. It is interesting to see that the rotated instances help for the data augmentation. However, I presume that the instances in a highly cluttered or distant scene would not be that effective. Such instances would make unrealistic samples because they are partially observed in a particular viewing direction. Some analysis of the data effectiveness or performance gain versus the distribution of the distant objects would be interesting.
3. Please check the weakness section and answer the questions there too.


**Limitations:**

- The paper does not address the limitation of the proposed approach. The concern about the computational overhead is clarified and stated. If it has drawbacks in any facts, the limitation section needs to describe them.

**Strengths And Weaknesses:**

**Strengths**
1. The paper comprehensively summarizes the related work in the point cloud augmentation field. The paper is self-contained, so the readers readily follow the problem and the recent advances.
2. The paper is straightforward to understand. I enjoyed reading the paper. The idea of mixing the 3D scenes is already known, but mixing on the polar coordinate seems to be very effective on the cylindrical LiDAR point clouds.
3. The approach shows compelling results on the SemanticKITTI [1], nuScenes-lidarseg [2], SemanticPOSS [22], SynLidar datasets. The proposed augmentation approach is applied with recent 3D semantic segmentation networks, such as MinkNet [8], SPVCNN [30], RandLA-Net [15], Cylinder3D [41]. For the task of 3D detection, PointPillar [18], Second [36], CenterNet [10] are applied. The gain is clear, and it outperforms other baselines.
4. The approach shows the effectiveness of the unsupervised domain adaptation as well. Since the approach can mix labeled source data and unlabeled target data, the approach can be readily applied to the various combinations of the domains.
5. The proposed augmentation approach improves data efficiency, as demonstrated in the experiment section. With the small amount of 3D scans, the PolarMix can produce a similar performance.
6. The paper explains the proposed idea in detail. The supportive figures (such as Figures 1 and 2) help to understand the approach better.

**Weakness**
1. The approach is a simple extension of the idea of the mix of 3D scenes and rotating bounding boxes (limited to the azimuth angles). The idea is not entirely new, and the target domain is limited to the cylindrical LiDAR datasets, not the general 3D scenes. However, the cylindrical LiDAR domain is one of the exciting domains for the task of intelligent mobility systems, and the custom design of 3D detection for the cylindrical LiDAR data also forms a research field. Therefore, I think it is not a critical weakness.
2. It is unclear how the baseline approaches for the semantic segmentation are selected. In addition to the CutMix [37], Copy-Paste [13], Mix3D [21], there are possible options to be applied. Similarly, approaches to augment the 3D detection task, such as GT-Aug [36, 12], CutMix, or approaches of [6, 11, 13] could be applied. The paper needs some clarification on how the baseline approaches were selected.
3. It is recommended to indicate the computational overhead when the proposed approach is applied. For instance, when training MinkNet, compared with the vanilla MinkNet training, what percentage of the total time is added to use PolarMix? Depending on the additional computation burden, the baseline approaches could be reevaluated.

---

> ### Author Response · Authors · 2022-08-02
> **Thanks for your appreciation of our proposed approach, impressive experimental results, as well as the clearness of presentation**
>
> Thanks for your appreciation of our proposed approach, impressive experimental results, as well as the clearness of presentation. Below please find our responses regarding your concerns.
>
> Q1: How the baseline approaches are selected?
>
> - Thanks for the insightful comment! The baseline in this study is very limited as data augmentation for LiDAR semantic segmentation is a relatively under-explored task (as stated in Lines 80-102 in Related Works). To address this constraint, we selected the highly-related mixing-based methods including Cut-Mix and Copy-paste in 2D vision and the pioneering work Mix3D for point cloud augmentation. We will further clarify the baseline issue in the experiment part in the updated manuscript.
>
> ---
>
> Q2: Additional training time of MinkUNet after using PolarMix?
>
> - Thank you for the great suggestion! We would clarify that PolarMix introduces less than 1\% of extra training time only as compared with the vanilla MinkNet. The superb efficiency is largely attributed to two major factors:
>   - The swapping and rotate-pasting process in PolarMix can be achieved by simple dot products, slicing, and concatenation which are extremely efficient (Line 135-137).
>   - Most LiDAR-based models sub-sample a fixed maximum number of points as input (e.g., 80k for SemanticKITTI in vanilla MinkNet and SPVCNN training). Hence, the network processes a similar amount of points (and takes little extra training time) although the augmentation introduces more points by cutting and mixing multiple copies of instances across scans.
>
> ---
>
> Q3: Mixing more scans?
>
> - We conducted the suggested experiments by increasing the mixed LiDAR scans and benchmarking them with no mixing. The experiments were conducted with SPVCNN that is trained with sequence 00 of SemanticKITTI. As Table I shows, mixing two scans produces clearly the best performance. We examined the mixed data and found that mixing more scans introduces more hardly distinguishable objects. The experimental results are well in line with other mixing-based augmentation works [7, 32, 38, 21].
>
>  Table I (with newly conducted experiments): Varying number of mixed scans by PolarMix. 'no mixing' represents the vanilla training without augmentation of PolarMix.
>
> | \#Scans | no mixing (baseline)| 2 | 3 | 4 |
> | :----: | :----: | :----: | :----: | :----: |
> | mIoU | 48.9 | **54.8** | 52.2 | 51.3 |
>
> ---
>
> Q4: More analysis of the performance gains versus the distribution of depth?
>
> - Thank you for the constructive suggestion! We evaluated the performance gains of PolarMix in recognizing points across different depths. Specifically, we split points of different ranges of depth and report segmentation performances of MinkNet over each split. The experimental results, as shown in Table J, are aligned with your thinking - the performance gains decrease with the increase of depth. We will include this experiment in the updated manuscript/appendix.
>
> Table J (with newly conducted experiments): Segmentation performances of MinkNet over points in different ranges of depth.
>
> | Depth (in meter) | [0,20) | [20, 40) | [40, 60) | [60, 80] |
> | :----- | :----- | :----- | :----- | :----- |
> | MinkUNet(baseline) | 61.0 | 48.6 | 29.6 | 47.7 |
> |+PolarMix | 66.8(+5.8) | 54.8(+6.2) | 34.0(+4.4) | 48.8(+1.1) |

---

### Official Review · Reviewer_cqy3 · 2022-07-12

**Rating:** 4
**Confidence:** 4
**Soundness:** 2 fair
**Presentation:** 2 fair
**Contribution:** 2 fair

**Summary:**

This paper proposes an augmentation method for point clouds, especially captured in road environments. The proposed augmentation method is to crop, cut, and mix two 3D scans at both scene-level and instance-level. This concept can be extended to unsupervised domain adaptation (UDA) by fusing source domain point cloud (known labels) and target domain point cloud (unknown labels). Series of experiments demonstrate its superiority in comparison with other point cloud augmentation methods.

**Questions:**

1. Intuition

1-1. I understand that this method effectively improves the quality of several target tasks. However, what is the fundamental intuition behind the proposed data augmentation? Why do such strategies bring performance improvement? While authors demonstrate lots of experiments that consistently increase the accuracy, I cannot expect why such an augmentation scheme is effective and novel.

1-2. What kinds of wrong estimations are fixed after applying the proposed augmentation strategy, PolarMix? For instance, is the estimation of the far-away points more accurate? Or, is this method robust to the density of 3D scans? Even though I read the additional results in the supplementary material, I cannot find any deep analysis of this issue.

2. Experiments

2-1. What if we fuse more than 3 scans?

2-2. What if there is a misalignment after data augmentation? In Fig2 of the manuscript, PolarMix crops point clouds in a certain range of azimuth. It means that the origins of the two-point clouds are aligned. I wonder why such assumption or alignment should be required for this augmentation? Why should we crop point clouds along the azimuth axis? Do you have any reason? (This is also related to Q.1-1)

2-3. Lack of comparison against recent UDA methods in Table5 of the manuscript.
In recent days, there are papers about UDA for 3D semantic segmentation, such as xMUDA [H]. In my opinion, the limited demonstration in Table5 of the manuscript does not fully support the authors' claims about the UDA setup. Technically speaking, PolarMix can be applicable to UDA and I agree with the authors' claims. However, I cannot judge whether this augmentation scheme is much more effective for the UDA for the 3D semantic segmentation task in comparison with previous methods [H]. Accordingly, the authors overclaim their novelty in UDA for the 3D semantic segmentation due to their limited comparisons.

3. Writing
3-1. (L208) As far as my understanding, MinkUNet also consists of sparse convolutional layers as SPVCNN did. However, the sentence can mislead readers as MinkUNet processes dense voxel-based architectures in contrast to SPVCNN.

3-2. I wonder why the authors did not present their qualitative results in the main manuscripts. Not just the quantitative results, qualitative results can help readers' understanding of this work. I recommend authors to revise the paper if it is accepted.
(I found the results in the supplementary materials. However, for completeness, I still think that the qualitative results should be listed in the main manuscripts.)

> If authors relieve my worries, I am willing to change my score.

Reference
- [H] xMUDA: Cross-Modal Unsupervised Domain Adaptation for 3D Semantic Segmentation, CVPR 2020


**Limitations:**

1. While authors only focus on the LiDAR scans that are captured in the road environments, there are other types of point cloud datasets, such as S3DIS [I] or ScanNet [J]. In my understanding, this method is not applicable to the indoor point set datasets. If so, authors should have included this problem as their limitation.

Reference
- [I] 3d semantic parsing of large-scale indoor spaces, ICCV 2016
- [J] Scannet: Richly-annotated 3d reconstructions of indoor scenes, CVPR 2017


**Strengths And Weaknesses:**

1. Strength
- Simple and straightforward ways of augmenting point clouds.
- Potential extension to UDA tasks.
- Consistent performance improvements in various tasks.

2. Weakness
- Lack of analysis and intuition behind such a design.
Overall, the authors present extensive experiments to demonstrate the superiority of the proposed data augmentation scheme. However, I wonder why such data augmentation is helpful for point-based recognition tasks. For instance, is PolarMix effective in making the networks more robust to point cloud density? or point cloud noise? The lack of such analysis makes me suspicious of the novelty of this work. Currently, this is the dominant worrying point of this paper.

---

> ### Author Response · Authors · 2022-08-02
> **Thanks for your acknowledgment of the value of our technical method and impressive experimental results across various tasks.**
>
> Thanks for your acknowledgment of the value of our technical method and impressive experimental results across various tasks. Below please find our responses regarding your concerns.
>
> Q1-1: Why PolarMix is novel and brings significant performance improvements across several target tasks?
>
> - As described in Lines 48-56, PolarMix works excellently because it enriches the diversity of LiDAR point data yet ensures their fidelity concurrently. As a comparison, most existing methods such as Mix3D [21] can improve the data diversity as well but they impair the data fidelity which affects their effectiveness.
> - In addition, Lines 39-46 provide detailed analysis and discussion. We copy and summarize the relevant text below for your quick reference.
>   - As compared with other augmentation methods (e.g. CutMix [37], Copy-paste [13], Mix3D [21]), PolarMix preserves unique properties of augmented LiDAR data by mixing points along azimuth direction, i.e. 1) partial visibility which means objects in LiDAR scans are incomplete where only object sides facing the LiDAR sensor are scanned with points as illustrated in Fig. 1(a); 2) point density decreases with the increase of depth as illustrated in Fig. 1(b). As a result, the augmented LiDAR scans of PolarMix are more realistic.
> - The two favorite properties have been experimentally verified in Section 4.1.2, where extensive experiments show that PolarMix outperforms the state-of-the-art point cloud augmentation methods consistently with clear margins.
>
> ---
>
> Q1-2: More deep analysis for PolarMix?
>
> - Thanks for your constructive comments. We conducted the suggested new experimental analysis which will be included in the updated manuscript. We show that **PolarMix increases the recognition robustness in both spatial locations and scene layout**.
> - Firstly, we rotate instances in the testing LiDAR scans and report segmentation performances of MinkNet w/ or w/o using PolarMix, which evaluates how models recognize instances appearing in different spatial locations.
>   - Table E below shows experimental results. It can be seen that the baseline performance drops significantly while rotating instances by different angles. This is largely because the baseline is very sensitive to the spatial location of instances that is often severely imbalanced in most existing datasets (due to LiDAR data collection and annotation constraints).
>   - As a comparison, PolarMix is robust to the instance spatial location without much performance drop. The experimental results show that PolarMix effectively improves the generalization of the trained LiDAR model (with respect to the instance spatial location) by generating lots of training samples at different spatial locations.
> - We then swap sectors of two testing scans with different angles, which generates new testing LiDAR scans with different layouts of road scenes. Similarly, we report segmentation performances of MinkNet w/ or w/o using PolarMix.
>   - The results are summarized in Table F below. We can observe that the baseline performance drops significantly while swapping sectors in the testing set with different angular ranges. However, the models trained with PolarMix are more robust with much less performance drop, indicating that PolarMix improves the robustness of LiDAR models with respect to the scene layout effectively.
>
> Table E (with newly conducted experiments): Segmentation with MinkUNet over the validation set of SemanticKITTI with rotated instances. PolarMix improves the robustness of the baseline clearly with respect to the angular variations of instances (i.e. spatial location variations).
>
> | method | $0^\circ$ | $45^\circ$ | $90^\circ$ | $135^\circ$ | $180^\circ$ |
> | :----- | :-------: | :-------: | :-------: | :-------: | :-------: |
> | baseline | 58.9 | 58.0(-0.9) | 57.6(-1.3) | 57.9(-1.0) | 57.5 (-1.4) |
> | +PolarMix | 65.0 | 64.9(-0.1) | 64.9(-0.1) | 65.0(-0.0) | 64.8(-0.2) |
>
> Table F (with newly conducted experiments): Segmentation result of MinkUNet over validation set of SemanticKITTI. We swap sectors of testing LiDAR scans to diversify layouts of road scenes and report mIoU performances. PolarMix significantly increases the robustness of the segmentation model.
>
> | method | $0^\circ$ | $45^\circ$ | $90^\circ$ | $135^\circ$ | $180^\circ$ |
> | :----- | :-------: | :-------: | :-------: | :-------: | :-------: |
> | baseline | 58.9  | 58.0(-0.9) | 56.4(-2.5) | 56.6(-2.3) | 57.6(-1.3) |
> | +PolarMix | 65.0  | 64.5(-0.5)  | 64.5(-0.5)  | 64.2(-0.8)  | 64.4(-0.6) |

---

> > ### Author Response · Authors · 2022-08-02
> > **Continued from previous responses**
> >
> > Q2-1: Fusing more scans?
> >
> > - We conducted the suggested experiments by increasing the mixed LiDAR scans and benchmarking them with no mixing. The experiments were conducted with SPVCNN that is trained with sequence 00 of SemanticKITTI. As Table G shows, mixing two scans produces clearly the best performance. We examined the mixed data and found that mixing more scans introduces more hardly distinguishable objects. The experimental results are well in line with other mixing-based augmentation works [7, 32, 38, 21].
> >
> >  Table G (with newly conducted experiments): Varying number of mixed scans by PolarMix. 'no mixing' represents the vanilla training without augmentation of PolarMix.
> >
> > | \#Scans | no mixing (baseline)| 2 | 3 | 4 |
> > | :----: | :----: | :----: | :----: | :----: |
> > | mIoU | 48.9 | **54.8** | 52.2 | 51.3 |
> >
> > ---
> >
> > Q2-2(1): What if there are misalignment for mixing?
> >
> > - LiDAR data are captured in a local coordinate system and the origin of LiDAR scans is the LiDAR sensor, which means the misalignment of two scans is negligible.
> >
> >
> > Q2-2(2): Why cropping point clouds along the azimuth axis?
> >
> > - As responded in Q1, we crop LiDAR points along the azimuth direction for maintaining the fidelity of the augmented point cloud data and preserving two unique and important LiDAR data properties (Lines 39-56): 1) objects in LiDAR scans are incomplete where only object sides facing the LiDAR sensor are scanned with points as illustrated in Fig. 1(a) in our submitted manuscript; 2) point density decreases with the increase of depth as illustrated in Fig. 1(b) in our submitted manuscript.
> >
> > ---
> >
> > Q2-3: Lack of comparison against recent UDA methods?
> > -  We evaluated PolarMix over the latest and challenging UDA benchmarks including SynLiDAR$\rightarrow$SemanticKITTI and SynLiDAR$\rightarrow$SemanticPOSS. As Table 5 in our submitted manuscript shows, PolarMix outperforms the recent UDA method PCT (AAAI2022) by a large margin. We copied the results in Table 5 in our submitted manuscript for your quick reference, please find in Table H below.
> > - Thanks for sharing the xMUDA. We note that xMUDA is a multi-modal UDA method that consists of cross-modal and uni-modal UDA. We compare PolarMix with the uni-modal xMUDA since PolarMix is designed for the single-modal LiDAR data. As Table H shows, PolarMix clearly surpasses xMUDA in both benchmarks, indicating the effectiveness of PolarMix in mitigating domain gap of LiDAR point clouds. We will update manuscript and include the comparison with xMUDA.
> >
> > Table H: Experiments on unsupervised domain adaptation with SynLiDAR (as source) and SemanticKITTI and SemanticPOSS (as target). PolarMix achieves clearly the best semantic segmentation across both unsupervised domain adaptation setups.
> >
> > | Methods | Publication | SynLiDAR $\rightarrow$ SemanticKITTI  | SynLiDAR $\rightarrow$ SemanticPOSS |
> > | :------ | :-------: | :-------: | :-------: |
> > | Source Only | - | 20.4 | 20.1 |
> > | ADDA [31] | CVPR2017 | 22.8 | 24.9 |
> > | Ent-Min [33] | CVPR2019 | 25.5 | 25.5 |
> > | ST [43] | CVPR2019 | 26.5 | 27.1 |
> > | PCT [35] | AAAI2022 | 28.9 | 29.6 |
> > | xMUDA [H] (newly included) | CVPR2020 | 28.5 | 28.9 |
> > | PolarMix(Ours) | - |  **31.0** | **30.4** |
> >
> > ---
> >
> > Q3: Writing issues?
> > - Thanks for your suggestion. MinkUNet is a sparse voxel-based convolutional network while SPVCNN is a hybrid network with sparse voxel-based convolutional layers and point-based neural layers. We will revise relevant text to make it clearer.
> > - We agree that qualitative results are helpful for the understanding and we provided them in the supplementary material due to the finite spaces of the manuscript. We will include qualitative results in the updated manuscript.
> >
> > ---
> >
> > Q4: Limitation of PolarMix?
> > - PolarMix is specifically designed for outdoor LiDAR point cloud learning and is not applicable to the dense indoor point set datasets. We will make clear representations about it.

---

> > > ### Comment · Reviewer_cqy3 · 2022-08-03
> > > **Can authors specify the details of UDA experiments?**
> > >
> > > Thank you for the precious rebuttals. I have some further questions after reading the rebuttals.
> > >
> > > 4. Domain setup
> > >
> > > I checked the manuscript of xMUDA [H] and the proposed dataset setup (Table H of this rebuttal) is different from the one that xMUDA proposed. I also agree that there is a modality difference between xMUDA and PolarMix, which makes it difficult to exactly align the same experimental setup. However, still, it does not necessarily change the dataset setup.
> > >
> > > 4-1. Is this dataset setup widely used in other UDA papers?
> > >
> > > 4-2. Can authors further specify how they train the PolarMix in the Table H of this rebuttal? In L240-L242 of the manuscript, the authors use pseudo labels to train the network using target domain data. Did you naively treat pseudo labels as ground truth labels without any filtering process?
> > >
> > > 5. Revised manuscript?
> > >
> > > I downloaded the manuscript after reading the rebuttals but have no idea whether this is the 'revised' manuscript. Did the authors revise the manuscript regarding the reviewers' comments? Commonly, as an intermediate revision, the authors can colorize the modified sentences during the discussion period.

---

> > > > ### Author Response · Authors · 2022-08-03
> > > > **Thank you for further comments.**
> > > >
> > > > Thanks for your further comments. Below please find our clarifications.
> > > >
> > > > Q4-1: Is this dataset setup widely used in other UDA papers?
> > > >
> > > > Domain adaptive LiDAR point cloud segmentation is a relatively new research task. To the best of our knowledge, the dataset setup has been adopted in three studies [a, b, c]. Note we didn't review [b] and [c] as they were publicly accessible after the submission deadline of NeurIPS 2022. In addition, [b] focuses on UDA and [c] focuses on model adaptation. Our proposed PolarMix instead focuses on point cloud augmentation under fully supervised setups. The UDA experiment is just one possible extension of PolarMix as presented in our submission.
> > > >
> > > > [a] Transfer Learning from Synthetic to Real LiDAR Point Cloud for Semantic Segmentation. AAAI 2022.
> > > >
> > > > [b] CoSMix: Compositional Semantic Mix for Domain Adaptation in 3D LiDAR Segmentation. ECCV 2022.
> > > >
> > > > [c] GIPSO: Geometrically Informed Propagation for Online Adaptation in 3D LiDAR Segmentation. ECCV 2022.
> > > >
> > > > ---
> > > >
> > > > Q4-2: Implementation details?
> > > >
> > > > We adopted the typical self-training strategy which treats confident pseudo labels of target data as ground truth (i.e., the top 20\% of the highest prediction scores initially) and then applies them to re-train the segmentation networks. The whole training repeats the two processes for five rounds (2 epochs in each round) with a gradually increasing confidence threshold (adding 5\% after each round).
> > > >
> > > > ---
> > > >
> > > > Q5: Revised manuscript?
> > > >
> > > > The current manuscript is what we originally submitted and we didn't upload any revised versions after the submission deadline. The rebuttal contains several new experiments that we just conducted to respond to your comments. We plan to incorporate them into the manuscript/appendix to be updated later. Thanks.

---

> ### Author Response · Authors · 2022-08-08
> **To Reviewer cqy3:**
>
> Dear Reviewer cqy3:
>
> We thank you for the precious review time and valuable comments. We have provided corresponding responses and results, which we believe have covered your concerns. We hope to further discuss with you whether or not your concerns have been addressed. Please let us know if you still have any unclear parts of our work.
>
> Best regards,
> Authors

---

### Official Review · Reviewer_Hdgz · 2022-07-12

**Rating:** 7
**Confidence:** 4
**Soundness:** 3 good
**Presentation:** 3 good
**Contribution:** 3 good

**Summary:**

This paper presents a data augmentation method specifically designed for LiDAR point clouds. Two layers of data augmentation are introduced: scene level augmentation and instance level augmentation. The authors demonstrated the enhancement brought by this data augmentation method on various applications, and showed that the proposed data augmentation is superior than the state-of-the-art LiDAR data augmentation methods.


**Questions:**

Will the authors release code after publication?


**Limitations:**

This paper presented a neat idea and showed extensive experiments to justify its contribution. I do not see any strong limitations.

**Strengths And Weaknesses:**

+ The idea is simple, in a good way
+ The evaluation is thorough: the authors demonstrated three application, namely semantic segmentation, object detection, domain gap reduction
+ The proposed method outperforms the state-of-the-art LiDAR data augmentation methods

---

> ### Author Response · Authors · 2022-08-02
> **Thank you for your appreciation of the novelty and simplicity of our proposed methods, the clearness in presentation, and thorough evaluation across various tasks.**
>
> Thank you for your appreciation of the novelty and simplicity of our proposed methods, the clearness in presentation, and thorough evaluation across various tasks. Below please find our responses regarding your concerns.
>
> Q: Will the authors release code after publication?
>
> - Yes, we are committed to open-source research and will release our codes upon the acceptance of this work.

---

### Official Review · Reviewer_NzUE · 2022-07-13

**Rating:** 5
**Confidence:** 4
**Soundness:** 3 good
**Presentation:** 3 good
**Contribution:** 2 fair

**Summary:**

This paper presents a polar coordinate system-based data augmentation approach, PolarMix, for scanning LiDAR point cloud understanding. PolarMix generates augmented data by mixing two different scans with scene-swapping and instance rotate-pasting. Extensive experiments and ablation studies validate the performance gain by PolarMix for three tasks（semantic segmentation, object detection, domain adaption）on three datasets.

**Questions:**

Besides the detailed ablations studies in the work, I am also curious about the following questions:
- This work tried cutting and swapping points along azimuth directions, how about further subdividing it along inclination direction.
- Is that possible to mix more than two scans? For example,  generate a new can by mixing each quarter from 4 different scans.
- What do σ_1 and  σ_2 indicate (L. 185)?
- For the rotation angles of rotate-pasting, any reason about why choosing values as L.183-185 for three datasets?

**Limitations:**

The authors didn’t discuss the limitations of their research and potential negative societal impacts.

**Strengths And Weaknesses:**

Strengths:
- The paper is well-written and easy to follow.
- The augmentation strategy is beneficial not only for semantic segmentation tasks but also for object detection and domain adaption. I like the idea of mixing two domains’ data scan-wisely to bridge the domain gap.
- Since PolarMix is a DA strategy, generally it is model-agnostic and applicable to any model.

Weaknesses:
- The baselines for comparison of w/  and w/o PolarMix are not the latest ones. I am wondering how is the performance gain by DA of  PolarMix under some latest backbones with higher baseline performance, the improvement will still significant?
- The contribution is a little limited, considering the relevant work. PolarMix is kind of the combination of CutMix, Copy-Paste and Mix3D from the perspective of idea.
- At least one relevant work is not mentioned and compared for object detection task. [1]

[1]J. Fang, X. Zuo, D. Zhou, S. Jin, S. Wang and L. Zhang, LiDAR-Aug: A General Rendering-based Augmentation Framework for 3D Object Detection, CVPR2021

---

> ### Author Response · Authors · 2022-08-02
> **Thanks for your confirmation of the value of our proposed approach, impressive experimental results as well as clearness of presentation.**
>
> Thanks for your confirmation of the value of our proposed approach, impressive experimental results as well as clearness of presentation. Below please find our responses to your concerns.
>
> Q1: PolarMix on the latest baseline models?
>
>  - We would clarify that SPVCNN [30] and Cylinder3D [41] tested in our submitted manuscript are two of most advanced open-source segmentation networks for LiDAR point clouds. SPVCNN with conventional global augmentation ('SPVCNN+CGA' in Table 1 in our submitted manuscript) achieves competitive segmentation performance (mIoU of 60.7\% over validation set of SemanticKITTI) with very fast training speed (less than 1 day) while Cylinder3D achieves SOTA performance (mIoU of 65.9\% as reported in Table 3 in [41]) with relatively longer training time. Both networks are widely used in the 3D point cloud community. In addition, PolarMix significantly boosts SPVCNN and achieves a mIoU of 66.2\% as shown in table below (extracted from Table 1 in our submitted manuscript), indicating that it is indeed a SOTA augmentation method for LiDAR point cloud learning.
>
> | Method               | mIoU      |
> | :------------------- | :-------- |
> | Cylinder3D[41]       | 65.9 (copied from [41])      |
> | SPVCNN [30]  | 60.7 |
> | SPVCNN+PolarMix | **66.2** |
>
> ---
>
> Q2: Is our contribution limited?
>
>  - We would clarify that the proposed PolarMix is a new data augmentation method as compared with previous methods including CutMix [37], Copy-Paste [13] and Mix3D [21]: CutMix and Copy-Paste from 2D vision work in Cartesian coordinates system; Mix3D globally mixes 3D scenes in an out-of-context manner. All three methods disregard the LiDAR data properties and the augmented data thus lack of fidelity (Lines 39-46).
>  - Differently, PolarMix mixes points within the polar coordinate system which ensures the high-fidelity of the augmented point cloud data (Lines 48-50). Specifically, PolarMix consists of two innovative augmentation designs including scene-level swapping and instance-level copy-and-paste, both operating in the azimuth direction that is well aligned with the unique scanning mechanism of LiDAR sensors (Line 37-38) and preserving specific properties of LiDAR scans including partial visibility and density variation along the depth (Lines 42-47 and Figure 1).
>
> ---
>
> Q3: Refer to LiDAR-Aug (CVPR2021)?
>
>  - Thank you for suggesting this paper. We cited LiDAR-Aug in the related works of the manuscript [12] (Lines 93-94) and will provide a more detailed review of this work. We note that the two methods are essentially different and incomparable. LiDAR-Aug uses additional synthetic CAD models (e.g. cars and pedestrians) to augment LiDAR scans while our PolarMix does not use any additional data sources. In addition, LiDAR-Aug does not release CAD resources and code which makes reproduction and comparison almost impossible.
>
>
> ---
>
> Q4: Subdividing it along inclination direction?
>
>  - Thank you for your suggestion! We also considered mixing in the inclination direction in the stage of methodology design. After some preliminary studies, we found that subdividing in the inclination dimension either impairs LiDAR data fidelity or requires very complicated point rendering design to preserve LiDAR characteristics, e.g., for preserving partial visibility and density variation along the depth. As a comparison, cut and mix along the azimuth direction preserves point fidelity greatly and efficiently as it is well aligned with the point capturing and LiDAR scanning mechanism.
>
> ---
>
> Q5: Mixing more than two scans?
>
>  - Thanks for the suggestion. We conducted the suggested experiments by increasing the mixed LiDAR scans and benchmarking them without using PolarMix. The experiments were conducted with SPVCNN that is trained with sequence 00 of SemanticKITTI. As Table A shows, mixing two scans produces clearly the best performance. We examined the mixed data and found that mixing more scans introduces more hardly distinguishable objects. The experimental results are well in line with other mixing-based augmentation works [7, 32, 38, 21].
>
>  Table A (with newly conducted experiments): Varying number of mixed scans by PolarMix. 'no mixing' represents the vanilla training without augmentation of PolarMix.
>
> | \#Scans | no mixing (baseline)| 2 | 3 | 4 |
> | :----: | :----: | :----: | :----: | :----: |
> | mIoU | 48.9 | **54.8** | 52.2 | 51.3 |
>
> ---
>
> Q6: Writing issues?
> - Sorry for the typo. $\sigma_1$ and $\sigma_2$ in Line 185 should be $\delta_1$ and $\delta_2$ as in the Algorithm 1. We will revise them in the updated manuscript.
>
> ---
>
> Q7: Implementation details?
> - We randomly choose rotation angles from different angular ranges when generating multiple copies. There is no specific considerations for these parameters in Lines 183-185.

---

> ### Author Response · Authors · 2022-08-08
> **To Reviewer NzUE:**
>
> Dear Reviewer NzUE:
>
> We thank you for the precious review time and valuable comments. We have provided corresponding responses and results, which we believe have covered your concerns. We hope to further discuss with you whether or not your concerns have been addressed. Please let us know if you still have any unclear parts of our work.
>
> Best regards,
> Authors

---

### Meta-Review · Area_Chair_QLXm · 2022-08-27

**Recommendation:** Accept
**Confidence:** Certain

**Metareview:**

The proposed augmentation method for LIDAR Scans is to crop, cut, and mix two 3D scans at both scene-level and instance-level. The approach is not novel and a simple extension of the idea of the mix of 3D scenes and rotating bounding boxes. Another limitation is that the method cannot be applied to general 3D scenes. The reviews include A(7), WA(6), BA(5), two BR (4). After carefully checking out the rebuttals and discussions, I recommend the paper to be presented for the NeurIPS community.

**Award:**

No

---

### Decision · Program_Chairs · 2022-09-14

Accept